# CauSciBench: A Comprehensive Benchmark on End-to-End Causal Inference for Scientific Research

## Abstract

Large language models (LLMs) are showing increasingly promising progress in accelerating scientific research, yet their ability to facilitate causal inference for scientific discovery remains underexplored. We introduce CauSciBench, the first comprehensive benchmark to evaluate end-to-end causal inference for scientific research. CauSciBench comprises 367 evaluation tasks based on 100+ real-world research papers across 9 disciplines, augmented with synthetic scenarios and textbook examples. CauSciBench is the first to probe the complete causal analysis pipeline, from natural language problem formulation through variable selection and method choice to statistical model implementation and result interpretation—all without any intermediate hints. We evaluate 6 state-of-the-art models with various test-time scaling techniques, including Chain-of-Thought, Program-of-Thought, and ReAct prompting. The best-performing OpenAI-o3 with CoT prompting still attains a mean relative error (MRE) of 48.96% on problems derived from real-world research papers, highlighting a substantial gap between current model capabilities and the demands of research-level causal analysis. We call on the community to further explore new methods and rigorous evaluation for building agents that can reliably facilitate causal inference in the context of scientific research.

## 1 Introduction

Causal inference is fundamental to establishing cause-and-effect relationships in scientific discovery that guide critical decisions in disciplines such as social science (Imbens & Rubin, 2015), public health (Glass et al., 2013), and biomedicine (Kleinberg & Hripcsak, 2011). The integration of LLM-powered agents into scientific workflows (Zhang et al., 2025; Lu et al., 2025) has shown promising progress to automate complex causal inference procedures (Han et al., 2024; Wang et al., 2025), with broader implications to accelerate scientific research across diverse disciplines (Kiciman et al., 2024).

Evaluating agentic capabilities of frontier language models in causal reasoning poses unique challenges, as causal inference usually involves unobservable counterfactual outcomes (Holland, 1986) and demands mastery of sophisticated methodological frameworks. Existing approaches often presuppose that users can correctly specify causal problems and choose suitable methods (Liu et al., 2024b; Chen et al., 2025a), which may not reflect the full complexity of real-world research.

Existing benchmarks are largely fragmented in assessing separate aspects of causal reasoning. Text-based approaches primarily assess commonsense causal understanding (Romanou et al., 2023; Nie et al., 2023; Chen et al., 2024b; Cui et al., 2024) or formal reasoning (Jin et al., 2023; 2024; Chen et al., 2024a). On the other hand, implementation-based benchmarks like QRData (Liu et al., 2024b) assess the execution of causal inference methods on tabular datasets, but do not fully evaluate the formulation of problems from natural language descriptions.

To bridge these gaps, we present CauSciBench, a comprehensive benchmark designed to systematically evaluate end-to-end causal inference capabilities from problem formulation and variable selection to method choice, estimation, and interpretation. Our work makes three key contributions:

**1. End-to-end Task Reflecting Research Demand.** CauSciBench is the first benchmark that requires models to perform the complete pipeline of causal inference: choosing treatment/outcome/con-

| Benchmark | End-to-End Causal Analysis | Intermediate Evaluation | Data + Context Understanding | Sources | Answer Format | # Queries |
|---|---|---|---|---|---|---|
| RealCause (Neal et al., 2021) | ✗ | ✗ | ✗ | 3 Datasets + Semi-synthetic Scenarios | Point Estimate | 1569 [1] |
| QRData (Liu et al., 2024a) | ✗ | ∼ | ✓ | 5 Datasets + 3 Textbooks | Freeform QA | 411 |
| DiscoveryBench (Majumder et al., 2024) | ✗ | ✗ | ✓ | 26 Datasets + Synthetic Scenarios | Freeform QA | 239 |
| BLADE (Gu et al., 2024) | ✗ | ✓ | ✓ | 12 Datasets | Code + Freeform QA | 12 |
| **CauSciBench** | ✓ | ✓ | ✓ | 100 Datasets + 2 Textbooks + Synthetic Scenarios | Point Estimate + Causal Components | **367** |

Table 1: Comparison of CauSciBench with related benchmark datasets for causal inference.
✓= Yes, ✗= No, ∼= Partial. **End-to-End Causal Analysis** indicates whether the benchmark evaluates the full pipeline of causal inference; **Intermediate Evaluation** captures whether the benchmark supports evaluation of intermediate steps; **Data + Context Understanding** assesses whether the benchmark requires models to interpret the relationship between the data variables and the background information.

founders, selecting appropriate identification strategies and estimation methods, implementing them, and finally interpreting results to conclude a given research problem.

**2. Hybrid Design with Real-Synthetic Comparison.** We combine three complementary data sources spanning real-world research problems, synthetic scenarios with user-defined ground truth, and adapted textbook examples to balance question validity with highly diverse problem sets. This design makes it possible to diagnose whether failures arise from implementation or from difficulties in handling the complexity of research problem descriptions.

**3. Vulnerability-Aware Automated Evaluation Pipeline.** To disentangle key vulnerabilities from our evaluation pipeline, we implement a fully automated evaluation capable of pinpointing key vulnerabilities in the causal inference pipeline, which usually boils down to problematic method selection or implementation. We further evaluate a wide range of frontier models and show that further effort is needed to reliably integrate LLM-powered agents into a research-level causal inference pipeline.

## 2 RELATED WORK

**LLM Benchmarks on Data-Driven Analysis** Early benchmarks primarily evaluate LLMs' ability to generate code for data visualization and pattern analysis (Yin et al., 2022; Lai et al., 2023; Li et al., 2024), with some extending to statistical reasoning for data-driven answers (Hu et al., 2024; Wu et al., 2024; Jing et al., 2025). More recent efforts target specialized domains such as machine learning (Huang et al., 2024; Nathani et al., 2025), biology (Laurent et al., 2024), and natural sciences (Chen et al., 2025c). However, most previous work has very limited coverage of social science, despite the central role of data-driven empirical analysis. Recent benchmarks such as BLADE (Gu et al., 2024) and DiscoveryBench (Majumder et al., 2024) introduce open-ended social science problems, but emphasize hypothesis validation with general data science tools rather than causal analysis. In contrast, CauSciBench directly evaluates LLMs' ability to perform rigorous causality-driven data analysis across diverse disciplines.

---

[1]Catesian product of 3 datasets, 4 estimators, 15 ML models, and 10 different hyperparameter settings

**LLM Benchmarks on Causality** Various benchmarks have emerged to evaluate LLMs' causal reasoning (Jin et al., 2024; Romanou et al., 2023; Nie et al., 2023; Chen et al., 2024a; Tu et al., 2024) and counterfactual reasoning (Chen et al., 2023; 2024b; Jin et al., 2023; Chen et al., 2025b). However, these benchmarks primarily test inference of causal relationships from natural language, rather than engaging with data. Likewise, data-driven causal benchmarks focus on causal discovery (Chevalley et al., 2023; Cheng et al., 2023; Zhou et al., 2024), where the task is to learn causal graphs from data. Our focus, on the other hand, is on **causal effect estimation**, where the goal is to quantify the effect of one variable on another using available data. Related works on causal effect estimation include QRData (Liu et al., 2024b), which evaluates whether models can implement user-specified causal inference methods, and RealCause (Neal et al., 2021), which focuses on evaluating different estimators. However, neither assesses the ability of models to autonomously identify appropriate variables and methods for estimating effects. CausalBench (Wang, 2024) consists of causal effect estimation tasks. However, these mostly involve synthetic scenarios and apply graph-based methods, such as front-door and back-door methods.

In contrast, CauSciBench predominantly focuses on examples using the Potential Outcomes Framework (Rubin, 2005), which is widely used in empirical research across social sciences, epidemiology, and bio-medicine. Moreover, CauSciBench tests the ability of LLMs to navigate the complete causal inference pipeline: identifying appropriate treatment and outcome variables, selecting and implementing suitable estimation methods, and providing meaningful interpretation of causal effects.

**LLM Agents for Causal Inference** The use case for LLM-powered agents has evolved from general machine learning and statistical analysis (Guo et al., 2024) to causal agents and foundation models, as exemplified by CausalAgent (Han et al., 2024), Causal-PFN Ma et al. (2025), CausalCoPilot (Wang et al., 2025), LLM4Causal (Jiang et al., 2024), and MAC (Le et al., 2025). While the development of these agents showcases the promising potential of LLM for science, they are largely using synthetic scenarios and/or focus on causal discovery tasks, leaving the challenges of real-world causal estimation under-evaluated. CauSciBench fills this gap by offering a systematic framework to evaluate agentic capabilities in scientific workflows that mirrors how practitioners leverage causal inference methods to tackle real-world research questions using available data.

## 3 BUILDING A COMPREHENSIVE BENCHMARK FROM REAL RESEARCH

We compile our dataset from three main sources, as illustrated in Figure 1. We introduce the dataset compilation steps below.

**Source 1: Real-World Research Papers** Statistical experts annotate research publications with open-sourced datasets from a wide range of disciplines, such as economics, public health, and political science, from sources like **Harvard Dataverse, Yale ISPS Data Archive**, and **R packages**. For each study, we curate a summary of the dataset, including variable descriptions, data collection, and research purposes. Next, we formulate causal queries answered in the study; if more than one causal treatment or outcome is present in the same paper, we curate multiple queries accordingly. We manually replicate the causal estimation results from each reference paper in Python to verify study replicability. To ensure query quality, two causality experts independently review each query across two validation rounds, with approval requiring consensus on satisfactory quality. The experts verify that dataset descriptions do not mention the underlying causal method and the findings of the study, and that queries avoid reference to model variables (e.g., treatment, outcome).

**Source 2: Synthetic Scenarios by Scalable Synthesis Framework** We randomly select the true causal effect $\tau$ in the range $(1, 10)$. Continuous covariates are drawn from a normal distribution, while binary covariates and treatment assignments (for binary treatment settings) are generated from a binomial distribution. The outcome $Y$ is determined by the model specification. For example, for a randomized trial:

$$Y = \alpha + X\vec{\theta} + \tau T + \epsilon, \tag{1}$$

where $\epsilon \sim \mathcal{N}(0, 1)$ is the error term, $\vec{\theta} \sim \mathcal{N}(u, kI)$, and $\alpha$ is the intercept. Here, $X$ denotes the covariates and $T$ is the treatment variable. We prompt GPT-4o to synthesize diverse plausible scenarios explaining how and why the data have been collected. We also require the evaluated LLMs to produce dataset metadata such as headings and descriptions for covariates, treatment variables, and outcomes. This approach improves the diversity of our synthetic datasets and allows us to test the consistency of model performance in synthetic scenarios vs. real-world research paper-based

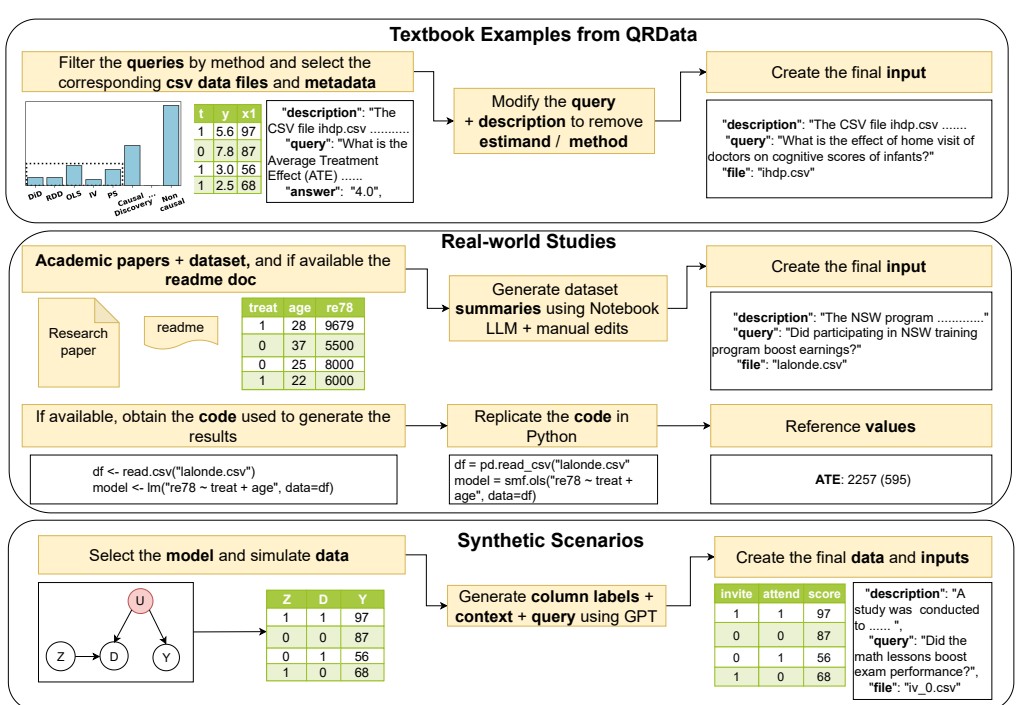

Figure 1: Illustration for building CauSciBench using 3 sources: QRData, Real-World Papers, and Synthetic Scenarios.

questions. The prompt template used for this task, as well as an example of the context and the associated query generated by an LLM, is provided in Appendix F.

**Source 3: Textbook-Based Datasets with Refinement** QRData (Liu et al., 2024c) contains causal inference tasks from textbooks (Alves, 2022; Imai, 2018). We refine QRData to only use queries with numerical answers. While QRData specifies the inference method or estimand, we carefully remove any explicit references to estimation techniques or causal effect measures since our focus is on end-to-end causal inference, which requires autonomous method/variable selection. As an example, the query related to the IHDP dataset (Hill, 2011): *What is the Average Treatment Effect (ATE) of the dataset?* becomes *What is the effect of home visits by doctors on cognitive scores of infants?*

**Contamination Concerns and Further Application** It's worth noting that our annotation framework can leverage the natural temporal structure of research publications to probe contamination patterns by evaluating questions synthesized from papers released before vs. after the model training cutoff. We believe it's a meaningful direction for future work that falls outside of the scope of this paper, with the main focus on introducing this dataset. We also recognize that any empirically grounded synthesis might raise concerns about data contamination. We will later show that despite such concerns, our evaluation results revealed how models have not exhibited strong memorization of the correct method in respective papers, which is a primary reason for model failure on our task.

### 3.1 CURATION PRINCIPLES AND CAUSAL INFERENCE METHODS

The core task we test is causal effect estimation with an appropriate method and variables. Each inference method is associated with a specific estimand and relies on particular assumptions for validity. In our benchmark, we consider widely used causal inference methods: regression discontinuity design (RDD) (Imbens & Lemieux, 2008), instrumental variables (IV) (Imbens, 2014), ordinary least squares (OLS) (Cunningham, 2021; Huntington-Klein, 2021), difference-in-differences (DiD) (Roth et al., 2023), matching methods (Stuart, 2010), propensity score-based methods (PS) (Rosenbaum

& Rubin, 1983; Austin, 2011), generalized linear models (GLMs) (Breen et al., 2018) as well as backdoor and frontdoor adjustment (Pearl, 2009).

## 3.2 STRUCTURE AND EXAMPLE OF A DATA POINT IN CAUSCIBENCH

Our goal is to evaluate LLMs on end-to-end causal analysis. This involves: (i) framing the causal estimation problem by selecting appropriate treatment and outcome variables with the correct estimand (target causal quantity), (ii) assessing whether the estimand can be identified and measured from the provided dataset, (iii) formulating and implementing the correct statistical model, and (iv) extracting and interpreting the causal effect in the context of the query.

To this end, each benchmark instance consists of four core components; the ones we denote with [Input] serve as input information for the model, and the ones with [Output] serve as a checker to evaluate the model's performance.

- **[Input] Dataset**: The input dataset (experimental or observational) with explanation including variable definitions and background context.
- **[Input] Query**: The causal question involving the effect of one variable on another.
- **[Output] Causal Inference Method and Effect Estimate**: The expert-validated causal method and corresponding effect. This provides ground truth for evaluating method selection and implementation.
- **[Output] Model Variables**: Key variables including treatment, outcome, confounders (variables affecting both treatment and outcome), and method-specific variables (e.g., instruments for instrumental variables). These act as the ground truth variable values to see if LLMs can choose the right variables for the causal model.

In Figure 2, we provide a sample annotation based on Card (1993) and provide full details of annotation attributes as well as guidelines for our expert annotators in Appendix G.

## 3.3 DIVERSE DOMAIN AND METHOD DISTRIBUTION

Figure 3 presents the distribution of paper domains in our real paper-based subset and the distribution of estimation methods across all three subsets of CauSciBench. We aim to include a wide coverage of causal inference scenarios and methods to reflect the complexity of real-world scientific research, which makes CauSciBench suitable for evaluation across various scientific domains and methodological approaches.

## 4 EXPERIMENTS

### 4.1 PROMPTING STRATEGIES

We leverage several test-time scaling strategies, namely Direct Prompting, Chain of Thought (CoT), Program-of-Thought (PoT), and ReAct-based prompting. Our prompt templates build upon the work of Liu et al. (2024b). However, we adapt the prompt for end-to-end causal estimation. Similarly, we use the backbone LLM (the LLM powering the causal inference process) to parse the causal estimation results implemented in Python and extract key variables, including treatment variables, outcome variables, model-specific variables (such as instrumental variables), and statistical results. We provide the detailed prompts in Appendix E. While we acknowledge that there are many other test-time optimizations one can pursue, we choose to strike a balance between representative methods and reasonable budgets. We believe it is an interesting direction for future work to investigate how additional test-time scaling techniques perform on our benchmark.

**Direct Prompting (Brown et al., 2020)** We provide the model with comprehensive dataset information, including descriptions, summary statistics, column names, and types, alongside the causal question and available methodological options. The model must directly select a causal inference method and produce executable Python code. This approach tests the model's ability to make methodological decisions based solely on provided information, without explicit guidance on intermediate steps or implementation structure.

---

**Sample Query Based on Card (1993)**

**Paper Source:** Using geographic variation in college proximity to estimate the return to schooling (Card, 1993)

**Description:** The National Longitudinal Survey of Young Men (NLSYM) was conducted to collect data on demographics, education, and employment outcomes. Participants were tracked over time to study long-term patterns. The dataset used here comes from the 1976 wave of the survey. Variables in the dataset:

- lwage: log of wages
- educ: years of education
- exper: years of work experience
- black: 1 if Black, 0 otherwise
- south: 1 if lives in a southern state, 0 otherwise
- married: 1 if married, 0 otherwise
- smsa: 1 if living in a metropolitan area, 0 otherwise
- nearc4: 1 if there is a four-year college in the county, 0 otherwise

**Query:** What is the effect of education on earnings?
**Answer:** 0.132
**Standard Error:** 0.049
**Is Significant:** 1
**Method:** IV (Instrumental Variable)
**Instrument Variable:** nearc4
**Data File:** card_geographic.csv
**Reference in Paper:** Table 4 in Card (1993)
**Field:** Economics

---

Figure 2: Sample data point with color-coded treatment variable, outcome variable, and control covariates.

**Chain of Thought (CoT) (Wei et al., 2023)** We maintain the same input as the direct prompting approach, but break down the typical causal inference workflow into steps: First, we ask the model to reason about the treatment, outcome, and confounding variables, along with justifications for each variable choice. Next, we ask models to select an estimand and the corresponding inference method while reasoning about how the identification assumptions are satisfied. Finally, we ask the model to sketch the implementation steps, including pre-processing and variable selection from the dataset, followed by model implementation and output of the necessary values for result interpretation in the respective context.

**Program-of-Thought (PoT) (Chen et al., 2022)** We require the LLM to generate a complete Python program following a structured template with predefined comments that outline the causal inference workflow. The prompt includes explicit guidance for sequential steps: variable identification, inference method selection, and statistical estimation. This approach differs from Direct Prompting by providing a clear implementation structure in the form of concise comments, and from CoT by emphasizing systematic code execution over explicit methodological reasoning.

**ReAct (Yao et al., 2023)** We provide only the data frame and the query, and allow the LLM to generate an answer through an iterative process involving reasoning, acting, and observation. Rather than reasoning about and then implementing the entire plan of action all at once, the process is broken down by the model itself. First, it reasons about the next step (thought), implements it (acts), and analyzes the results to plan the next step (observation). This process is implemented iteratively until the agent finally settles on an answer.

For all prompting approaches, we incorporate an error correction mechanism. Upon encountering Python execution errors, we supply the LLM with the error information and allow it to rewrite the code. This retry process is permitted up to three attempts.

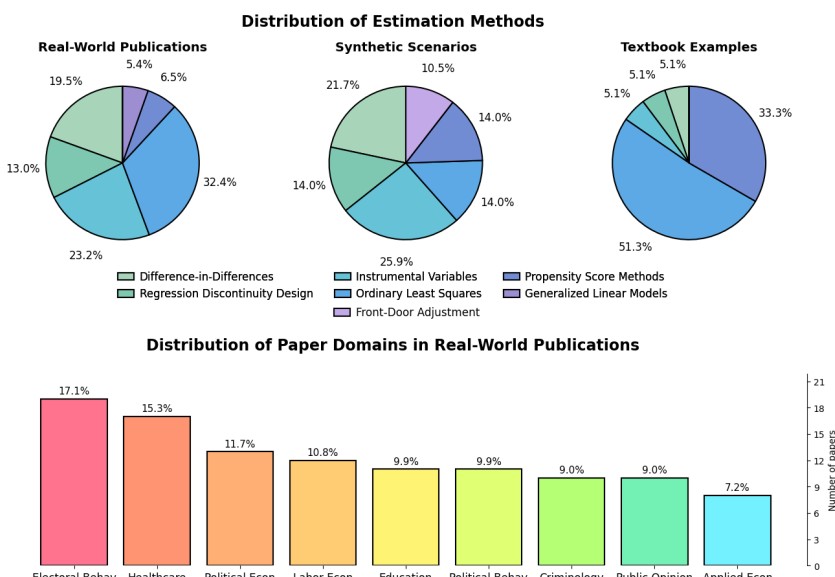

Figure 3: Distribution of paper domains in Real-world publications and estimation methods across the three dataset collections. The terms behavior and economics are abbreviated to Behav and Econ, respectively.

## 4.2 EVALUATION SETUP

**Python Libraries** For causal effect estimation, we use the DoWhy (Sharma & Kiciman, 2020; Blöbaum et al., 2024), linearmodels, (Sheppard et al., 2024), rdd (Magnusson, 2019), and statsmodels, (Seabold & Perktold, 2010) libraries. Similarly, for pre-processing and intermediate computations, we use numpy (Harris et al., 2020), pandas (pandas development team, 2020), and scikit-learn (Pedregosa et al., 2011).

**Metrics** Suppose $N$ denotes the total number of queries in the evaluation set. We evaluate all models using the following two metrics: (1) **Method Selection Accuracy (MSA)**: Percentage of queries where the selected method $\hat{m}_i$ matches the reference method $(m_i)$ MSA $= \frac{1}{N} \sum_{i=1}^{N} \mathbf{1}[\hat{m}_i = m_i] \times 100\%$. (2) **Mean Relative Error (MRE)**: Average relative error between predicted causal effects $(\hat{\tau}_i)$ and reference values $(\tau_i)$: MRE $= \frac{1}{N} \sum_{i=1}^{N} \min\left(\frac{|\hat{\tau}_i - \tau_i|}{|\tau_i|}, 1\right) \times 100\%$. To reduce the impact of outliers, relative error is capped at 100% per query.

## 4.3 RESULTS AND DISCUSSION

We tested 7 frontier models from leading model families, including OpenAI, Gemini, Grok, and Qwen. Table 2 shows the method-selection accuracy and relative errors (MRE) of causal effect estimates under pass@1. While it's possible to perform pass@k, we followed the best practice of previous work in this field and struck a balance between model coverage with budget considerations.

**Causal estimation from real data is challenging.** Method selection accuracies for real datasets consistently underperform synthetic and textbook datasets across all models and prompting strategies, typically ranging from 35-70% with mean relative errors exceeding 60%. While synthetic datasets benefit from controlled generation and textbook datasets from extensive preprocessing for pedagogy, real-world data presents greater complexity through more variables, higher noise levels, and a lack of preprocessing. These factors complicate both method and variable selection, with methodological errors cascading through the causal inference pipeline to amplify estimation errors.

**Wrong methods directly amplify estimation errors.** Table 4 in the appendix shows that incorrect method selection is the primary driver of causal inference failures, yielding substantially higher MRE across nearly all settings. This effect intensifies with dataset complexity, particularly for real-world data. Textbook-based dataset is an exception. This is because most misclassifications

| Dataset | Model | Method Accuracy (↑) | | | | Mean Rel. Error (↓) | | | |
|---|---|---|---|---|---|---|---|---|---|
| | | Direct | CoT | PoT | ReAct | Direct | CoT | PoT | ReAct |
| Real | Gemini-2.5-Flash-Lite | 42.86 | 51.63 | 51.91 | 54.34 | 67.64 | 66.08 | 71.82 | 72.09 |
| | Grok-4-Fast | 74.05 | 73.37 | 65.95 | 67.21 | 59.02 | 58.44 | 58.36 | 66.77 |
| | GPT-5-mini | 69.40 | 70.95 | 68.31 | 70.32 | 59.05 | 55.78 | 59.50 | 53.56 |
| | GPT-4o-mini | 32.58 | 36.93 | 36.52 | 35.43 | 76.01 | 75.44 | 76.17 | 68.02 |
| | GPT-4o | 56.52 | 59.34 | 56.14 | 59.41 | 67.09 | 67.44 | 74.20 | 66.58 |
| | OpenAI-o3 | 70.83 | 77.17 | 68.33 | 72.60 | 52.56 | 48.96 | 63.10 | 66.48 |
| | Qwen3-Next-80B-Inst | 56.76 | 62.70 | 57.07 | 55.80 | 61.57 | 67.52 | 70.65 | 68.64 |
| Synthetic | Gemini-2.5-Flash-Lite | 79.43 | 84.17 | 80.42 | 62.86 | 45.25 | 34.05 | 39.27 | 48.34 |
| | Grok-4-Fast | 76.81 | 76.76 | 69.72 | 71.74 | 15.86 | 13.08 | 23.64 | 32.14 |
| | GPT-5-mini | 87.77 | 91.43 | 93.48 | 84.68 | 7.91 | 7.93 | 14.33 | 9.40 |
| | GPT-4o-mini | 15.38 | 24.48 | 27.34 | 23.78 | 21.78 | 26.41 | 30.74 | 25.29 |
| | GPT-4o | 70.63 | 83.10 | 80.14 | 65.49 | 28.82 | 28.03 | 25.46 | 21.95 |
| | OpenAI-o3 | 86.47 | 91.35 | 79.58 | 80.00 | 8.43 | 43.97 | 17.20 | 71.57 |
| | Qwen3-Next-80B-Inst | 72.86 | 75.00 | 80.14 | 65.44 | 22.89 | 33.18 | 34.42 | 34.82 |
| Textbook | Gemini-2.5-Flash-Lite | 61.54 | 76.32 | 71.79 | 82.05 | 40.43 | 35.82 | 38.80 | 49.23 |
| | Grok-4-Fast | 66.67 | 66.67 | 66.67 | 63.16 | 29.03 | 23.79 | 26.31 | 25.60 |
| | GPT-5-mini | 69.23 | 75.68 | 71.79 | 61.76 | 41.14 | 42.02 | 47.34 | 32.91 |
| | GPT-4o-mini | 51.28 | 62.16 | 60.53 | 54.05 | 36.51 | 31.21 | 37.43 | 26.29 |
| | GPT-4o | 61.54 | 66.67 | 66.67 | 61.54 | 43.72 | 42.33 | 26.76 | 33.58 |
| | OpenAI-o3 | 66.67 | 60.71 | 66.67 | 72.73 | 30.10 | 37.93 | 44.56 | 63.48 |
| | Qwen3-Next-80B-Inst | 74.36 | 76.92 | 84.62 | 74.36 | 35.55 | 43.74 | 42.25 | 38.60 |

Table 2: Comparison of method accuracy (↑) and mean relative error (↓) across datasets, models, and prompting strategies. **Bold** values indicate the best result in each column across all models. For each model, dark green indicates the highest method accuracy and dark red indicates the lowest relative error for a given model across the 4 prompting approaches.

involve choosing propensity score methods over regression / difference-in-means in the IHDP dataset (Hill, 2011), a randomized experiment where both methods yield similar results.

**Implementation failures persist despite correct method choice.** Even with appropriate method identification, substantial errors remain due to execution failures, as evidenced by persistently high relative errors in Table 2. These residual errors stem from inappropriate variable selection, model mis-specification, or algorithmic implementation mistakes. This aligns well with the findings of Liu et al. (2024b), where GPT-4 achieved only 58% implementation accuracy even when provided the correct model. Our benchmark presents a more demanding challenge by requiring both methodological selection and functional implementation without external hints, measuring genuine end-to-end performance that human scientists must perform in real-world research.

**Models systematically default to OLS estimation.** The confusion matrices in Figures 4 reveal that LLMs exhibit a pronounced bias toward Ordinary Least Squares (OLS) across all causal inference scenarios, regardless of the appropriate method. This tendency is particularly pronounced for smaller models, such as GPT-4o-mini. The overwhelming selection of OLS stems from several factors. OLS is simpler and easier to implement. Likewise, for most empirical papers, OLS is the baseline model. However, this bias is highly problematic for causal inference. Naive OLS often fails to address the effect of unobserved confounders. Hence, when possible, researchers use instrumental variables. Likewise, even when confounders are observed, naive OLS-based estimates exhibit low precision. Thus, practitioners often employ techniques like matching (Dehejia & Wahba, 2002).

**Prompting strategies show conditional effectiveness.** As shown in Table 2, no single prompting strategy consistently outperforms others across all settings. While CoT prompting generally improves model selection accuracy over direct prompting, it can also degrade performance for OpenAI-o3 on textbook data. PoT and ReAct prompting exhibit even more variability, excelling in specific scenarios while underperforming in others. Notably, ReAct achieves the best accuracy for some models on textbook data but shows the worst performance on synthetic data for the same models. Furthermore, prompting methods that maximize accuracy often fail to minimize relative error, suggesting a trade-off between these metrics. These findings indicate that the effectiveness of structured prompting

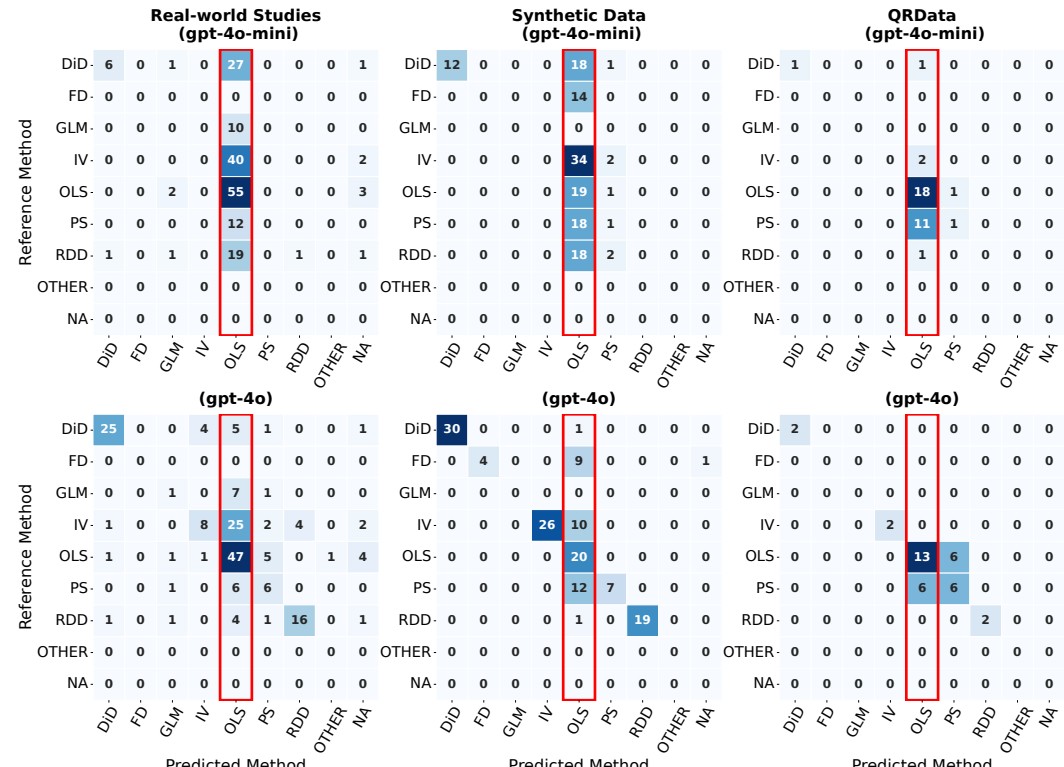

DiD: Difference-in-Differences | FD: Frontdoor Criterion | GLM: Generalized Linear Models
IV: Instrumental Variables | OLS: Ordinary Least Squares | PS: Propensity Score Methods(Matching + IPW)
RDD: Regression Discontinuity Design | OTHER: Methods Outside Benchmark | NA: Implementation Failure

Figure 4: Confusion matrix for method selection across the three datasets for GPT-4o-mini and GPT-4o, with results averaged across all prompting strategies (Direct, CoT, PoT, ReAct). The red boxes highlight the over-reliance of models on ordinary least squares (OLS). However, this over-reliance is reduced for larger models.

techniques depends heavily on model architecture, dataset characteristics, and target metrics, with implementation-oriented tasks potentially suffering from over-structured reasoning approaches, aligning with what Liu et al. (2024b) suggest. This underscores the need for task-specific and model-specific prompting selection rather than universal strategies.

## 5 CONCLUSION AND FUTURE WORK

We presented CauSciBench, the first comprehensive benchmark for evaluating LLMs' causal estimation capabilities in real-world scientific research. Our findings demonstrate that current LLMs exhibit systematic biases toward methodological oversimplification, such as defaulting to OLS estimation regardless of identification requirements, while simultaneously struggling with implementation accuracy even when their methodological reasoning proves sound. Moreover, the substantial performance gap between synthetic and real-world scenarios highlights critical limitations in existing approaches. Progress in LLM-based causal inference requires (1) high-fidelity datasets that capture the complexity of observational data, (2) methodological selection mechanisms beyond simple pattern matching, and (3) stronger integration of theoretical reasoning with practical implementation. Assessing these challenges is essential for developing LLMs' capability of reliably supporting causal inference and, ultimately, democratizing sophisticated causal analysis across disciplines.

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

## A  LLM USAGE, REPRODUCIBILITY AND ETHICAL STATEMENTS

**Declaration on LLM Usage**  We have strictly adhered to the ICLR policy of LLM Usage. We have used commercially available LLMs for proofreading to ensure this paper reads fluently without major grammatical errors.

**Reproducibility Statement**  We provide our complete codebase for reproduction in the supplementary material. We include detailed instructions on running our codebase in the README.md file of our supplementary material .zip file. Due to the file size limit on OpenReview, we have currently submitted the CSV files for the synthetic and textbook data collections. We plan to fully open-source our code, datasets (along with their usage licenses), and evaluation logs on HuggingFace after peer review.

We note that the output of language models can be highly non-deterministic by design, especially for reasoning models where the temperature parameter cannot be defined by users.

**Ethical Statement**  We use open-source datasets from public repositories including Harvard Dataverse, Yale ISPS Data Archive, and R libraries. The data and code in these repositories have been made available for research and sharing purposes under various licensing terms. We use the datasets as provided in the repositories without creating derivatives.

When we publicly release this benchmark, we will include complete licensing documentation to ensure the research community complies with original terms and provides proper attribution to dataset creators. We will provide clear links to original data sources and their respective licenses.

This work involves no primary data collection from human subjects. All real-world datasets are secondary data from previously published research. Synthetic data contains no perturbations of existing data and is simulated from pre-specified statistical models.

## B  LIMITATIONS

Our work has several limitations that warrant careful consideration. The expert-curated subset requires extensive manual curation to synthesize questions from long research papers, creating scalability constraints and potential annotation inconsistencies across different domains and methodological approaches. The results reported are based on pass@1 evaluation to balance budgetary constraints with broad model coverage, although a more comprehensive evaluation with pass@k would strengthen the generalizability of our findings. Our benchmark focuses primarily on the potential outcomes framework with limited coverage of Pearl's structural causal model approach, potentially under-representing certain causal reasoning paradigms prevalent in computer science and AI research. The synthetic data generation process, while systematic, may not fully capture the complexity and idiosyncrasies of real-world datasets, including missing data patterns, measurement error, and domain-specific confounding structures. Our evaluation metrics may not adequately capture the severity of estimation failures or provide sufficient granularity for understanding model performance across different effect sizes. The benchmark's temporal partitioning strategy for contamination detection assumes clear publication cutoffs, but pre-print availability and gradual knowledge diffusion may complicate contamination assessment. Additionally, our focus on English-language publications from primarily Western academic institutions may limit the cultural and methodological diversity of causal inference approaches represented in the benchmark. The binary treatment focus excludes important multi-valued and continuous treatment scenarios common in many scientific applications, while the emphasis on tabular data overlooks emerging applications of causal inference to text, images, and other high-dimensional data modalities. Finally, our evaluation framework does not assess crucial aspects of scientific causal inference, including assumption testing, sensitivity analysis, and the communication of uncertainty, which are essential for the responsible application of causal methods in scientific research.

## C  DATASET CURATION PROCESS

The dataset curation process of our work follows a three-stage methodology, designed to ensure high-quality benchmarks through rigorous, expert-curated papers.

- **Paper Selection** focuses on finding articles from diverse fields such as healthcare and economics that utilize established estimation methods, including OLS, DiD, RDD, IV, and propensity score methods. The selection criteria emphasized reproducibility and dataset complexity, where we prioritize papers with simpler and explicit approaches to causal estimation to work with current LLMs' preprocessing limitations. Furthermore, as we go through the replication process in future steps, we exclude papers that do not include a publicly accessible dataset with adequate data sharing licensing.

- **Core Information Extraction** follows paper selection, focusing on extracting the core information that practitioners require for a causal analysis, including treatment variables, outcomes, and non-causal natural language queries to avoid any methodological hints. Multiple questions per paper are permitted when the controls or outcomes differ meaningfully, maximizing the scientific value, while preventing analytical redundancy.

- **Quality Filtering** implements multi-layered expert inspection throughout the entire curation process. All curated datasets undergo replication verification, where experts replicate the estimation process in Python, and exclude all papers that fail to reproduce the original estimates within 10% error in around 50 lines of code. This process validates that the estimates in the paper are truly replicable with the given dataset and methods, so that should the LLM fail to replicate the results, the cause lies in the LLM's approach, and not the dataset or the paper's approach.

# D    SAMPLE QUESTIONS FROM EACH SOURCE

| **Real-World Publications** |
| --- |
| **Source: Cities as Lobbyists (Goldstein & You, 2017)** |
| **Domain: Economics** |
| **Natural Language Query:** How much does the money spent on lobbying increase the number of earmarks received? |
| **Method:** Instrumental Variables |
| **Treatment:** `ln_citylobby` (log of city lobbying spending) |
| **Instrument:** `direct_flight_dc` (1=direct flight to DC in 2007, 0=otherwise) |
| **Outcome:** `ln_earmark` (log of total earmarks 2008-2009) |
| **Other Variables:**  `state, pop_e, land_e, water_e, senior_e, student_e, ethnic_e, mincome_e, unemp_e, poverty_e, gini_e, city_propertytaxshare_e, city_intgovrevenueshare_e, city_airexp_e, houdem_e, ln_countylobby` |
| **Data:** Cities with population over 25,000, 2007-2009 panel |
| **Synthetic Dataset** |
| **Source: Cardiovascular Rehabilitation Program Effectiveness Study** |
| **Domain: Healthcare** |
| **Natural Language Query:** Does the new rehabilitation program help patients with cardiovascular diseases recover faster? |
| **Method:** Regression Discontinuity Design |
| **Treatment:** `treatment_received` (1=new program, 0=standard care) |
| **Running Variable:** `income_level` (threshold at 12 for eligibility) |
| **Outcome:** `recovery_time` (days to recovery) |
| **Other Variables:** `patient_age, health_index, smoking_status, obesity_status` |
| **Data:** Regional health department evaluation study |
| **Textbook Examples** |
| **Source: Effect of Cigarette Taxation on Consumption (Liu et al., 2024b)** |
| **Domain: Healthcare, Political Science** |
| **Natural Language Query:** Did Proposition 99 help reduce cigarette sales? |
| **Method:** Difference-in-Differences |
| **Treatment:** `california` (1=CA with Prop 99, 0=other states) |
| **Time:** `after_treatment` (1=post-1988, 0=pre-1988) |
| **Outcome:** `cigsale` (total cigarette sales) |
| **Other Variables:** `state, year, lnincome, beer, age15to24, retprice` |
| **Data:** 39 US states, 1970-2000 panel |

Table 3: Sample questions from each source pillar with the information regarding the paper that the LLM uses as context.

# E    PROMPT TEMPLATES

In this section, we present the templates for two of the baseline prompting strategies: **Direct Prompt** and **Chain of Thoughts (CoT) prompt**

---

**Direct Prompt**

You are an expert in statistics and causal reasoning. You will answer a causal question on a tabular dataset.

The dataset is located at `{self.dataset_path}`.

The dataset has the following description: `{self.dataset_description}`

To help you understand it, here is the result of `df.describe()`:

`{df_info}`

Here are the columns and their types:

`{columns_and_types}`

Here are the first 5 rows of the dataset:

`{df.head()}`

If there are fewer than 10 columns, here is the result of `df.cov()`:

`{(df.cov(numeric_only=True) if len(df.columns) < 10 else "Too many columns to compute covariance")}`

Here is the output of `df.isnull().sum(axis=0)`: `{nan_per_column}`

The causal question I would like you to answer is: `{self.query}`

Using the descriptions and information from the dataset, write Python code to build the causal inference model based on the method and variables you have selected, and compute the causal effect to answer the query. If you need to preprocess the data, please do so in the code.

Important: Only use these approved packages: pandas, numpy, scipy, scikit-learn (sklearn), statsmodels, dowhy, rdd (for regression discontinuity design), linearmodels, econml.

Here are some example methods; you can choose one from them:

- IPW (Inverse Probability Weighting): choose the right estimand (ATE/ATT/ATC), and compute the causal effect
- Linear regression with control variables: build a regression model with the treatment, outcome, and confounders/control variables, and compute the causal effects
- Instrumental variable: build an instrumental variable model, and compute the causal effects associated with the treatment variable
- Matching: choose the correct estimand (ATE/ATT/ATC), and match accordingly, and then compute the causal effects
- Difference-in-differences: build a difference-in-differences model, and output the coefficient of the variable of interest
- Regression discontinuity design: build a regression discontinuity design model, and output the coefficient of the variable of interest
- Linear regression / difference-in-means: either build a regression model consisting of the treatment and outcome variables, and compute the coefficient associated with the treatment variable or compute the difference in means across treatment and control groups
- Generalized linear models / GLM: build a GLM model, and output the coefficient of the variable of interest
- Frontdoor adjustment: build a causal graph, identify a mediator variable between the treatment and outcome, check for frontdoor criterion, and compute the causal effect using the frontdoor adjustment formula

Make sure the code prints the final results, including:

---

1. The causal effect (the value only)

2. The standard deviation (the value only)

3. The causal inference method that was used to compute the effect (the method name only)

4. The treatment variable (the variable name only)

5. The outcome variable (the variable name only)

6. The mediator variable (the variable name only if frontdoor adjustment was used)

7. RCT: True / False (NA if not sure; whether the data is from a randomized controlled trial or not)

8. The covariates / control variables that were used in the causal inference model (the variable names only)

9. Instrumental variable, if instrumental variable method was used (the variable name only)

10. Running variable, if regression discontinuity design was used (the variable name only)

11. Temporal variable, if difference-in-differences was used (the variable name only)

12. Results of statistical tests, if applicable

13. Brief explanation for model choice

14. The regression formula, if applicable.

If a variable is not applicable, print "NA" for it.

The code you output will be executed, and you will receive the output. Please make sure to output only one block of code, and make sure the code prints the result you are looking for at the end. Everything between your first code block: ```python and ``` will be executed. If there is an error, you will have several attempts to correct the code.

---

## Chain of Thoughts Prompt

You are an expert in causal inference. You will use a chain-of-thought approach to answer a causal question on a tabular dataset.

The dataset is located at {self.dataset_path}

The dataset has the following description: {self.dataset_description}

Here are the columns and their types: columns_and_types

Here is the statistical summary of the dataset: df.describe()

Here are the first 5 rows of the dataset: {df.head()}

If there are fewer than 10 columns, here is the result of df.cov():

{(df.cov(numeric_only=True) if len(df.columns) < 10
else "Too many columns to compute covariance")}

Here is the output of df.isnull().sum(axis=0): {nan_per_column}

The causal question I would like you to answer is: {self.query}

Let us approach this problem step by step.
Step 1. First, go through the dataset description and the columns and their types. Then, identify the treatment variable, the outcome variable, and the potential confounders. Explain your reasoning for choosing these variables. Remember, the dataset is located at: {self.dataset_path}.

Step 2. What would be the right estimand to consider for this problem? Then, choose the most appropriate method that can be used to estimate the causal effect. The available methods are:

- IPW (Inverse Probability Weighting): Choose the right estimand (ATE/ATT/ATC), and compute the causal effect

- Linear regression with control variables: Build a regression model with the treatment, outcome, and confounders/control variables, and compute the causal effects

- Instrumental variable: Build an instrumental variable model, and compute the causal effects associated with the treatment variable

- Matching: Choose the correct estimand (ATE/ATT/ATC), and match accordingly, and then compute the causal effects,

- Difference-in-differences: Build a difference-in-differences model, and output the coefficient of the variable of interest

- Regression discontinuity design: Build a regression discontinuity design model, and output the coefficient of the variable of interest

- Linear regression / difference-in-means: Either build a regression model consisting of the treatment and outcome variables, and compute the coefficient associated with the treatment variable or compute the difference in means across

- treatment and control groups

- Generalized linear models / GLM: Build a GLM model, and output the coefficient of the variable of interest,

- Frontdoor adjustment: Build a causal graph, identify a mediator variable between the treatment and outcome, check for frontdoor criterion, and compute the causal effect using the frontdoor adjustment formula

Explain why you chose the selected method, and how the data and its description support your choice. This means you should explain why the identification assumptions of the method are satisfied.

Step 3. Next, we will plan the implementation. Before writing the code, describe your implementation process. This includes:

1. Describing the necessary pre-processing steps.

2. How we will select the variables to use in the model?

Step 4. Finally, reflecting on the previous steps, write Python code to answer the causal question: {self.query}. Feel free to preprocess the data.

Important: Only use these approved packages: pandas, numpy, scipy, scikit-learn, statsmodels, dowhy, rdd, linearmodels, econml.

Use the methods from the above libraries to implement the method you chose. Be careful about implementation.

Make sure the code prints the final results, including:

1. The causal effect (the value only)

2. The standard deviation (the value only)

3. The causal inference method used (the method name only)

4. RCT: True / False / NA

5. The treatment variable

6. The outcome variable

7. The mediator variable (if applicable)

8. The covariates / control variables

9. Instrumental variable (if applicable)

10. Running variable (if applicable)

11. Temporal variable (if applicable)

12. Results of statistical tests, if applicable

13. Brief explanation for model choice

14. The regression formula, if applicable

If a variable is not applicable, print "NA" for it.

The code you write will be executed, and you will next analyze the output. To ease the process, please output one block of code, and make sure the code prints the key results and values. Everything between your first code block: ```python and ``` will be executed. If there is an error, you will have several attempts to correct the code. Hence, if there is an error, please fix it and re-run.

## Program of Thoughts Prompt

You are a causal inference expert. Your goal is to generate a causality-driven answer to the user query: {self.query} using the provided data.

The description and the query can be found below. Please analyze the input information and write a Python code that performs causal effect estimation.

You can use the following libraries: pandas, numpy, scipy, sklearn, statsmodels, dowhy, rdd, linearmodels, econml.

The format of the code should be:
```python
def causal_analysis():
    # import libraries
    # load data
    # identify treatment, outcome, confounders,
    # control variables (pre-treatment variables)
    # select appropriate causal method, and method-specific variables
    # estimate causal effect and standard error
    # print results (12 items listed below). This is important
    # return a dictionary containing the 12 items listed below
result = causal_analysis()
```

Available causal inference methods: IPW (Inverse Probability Weighting), Linear regression with control variables, Instrumental variable, Matching, Difference-in-Differences, Regression Discontinuity Design, Linear Regression/Difference-in-Means, Generalized linear models, Frontdoor adjustment.

Print the following 12 items in the code:
1. Causal effect
2. Standard error
3. Method name
4. RCT (True/False/NA)
5. Treatment variable
6. Outcome variable
7. Mediator variable
8. Control covariates used
9. Additional variable
10. Statistical test results
11. Model choice explanation
12. Regression formula (if applicable)

If a field is not applicable, print "NA".

Here is information about the data. Data Description: {self.dataset_description}

Dataset Location: {self.dataset_path}

Columns and types: {columns_and_types}

First 10 rows: `{df.head(10)}`

Missing values: `{nan_per_column}`

Likewise, the query is: `{self.query}`

Everything between your first code block: python and will be executed. If there is an error, you will have several attempts to correct the code.

## ReAct Prompt

Data Description: `{self.dataset_description}`

The dataset is located at `{self.dataset_path}`

You are a causal inference expert working with a pandas dataframe in Python. The name of the dataframe is 'df'

You should use the tools below to answer the causal question of interest:

'python_repl_ast': A Python shell. Use this to execute Python commands. Input should be a valid Python command.|

When using this tool, sometimes output is abbreviated - make sure it does not look abbreviated before using it in your answer.

Important: Only use these approved packages: pandas, numpy, scipy, scikit-learn (sklearn), statsmodels, dowhy, rdd, linearmodels, econml.

Use the following format

```
Question: The input question you must answer
Thought: Your thoughts on what to do next. You need to think carefully.
Action: The action to take, should be python_repl_ast
Action Input: The input to the action, should be the code to execute
Observation: The result of the action
...(this Thought/Action/Action Input/Observation can repeat N times)
Thought: I now know the final answer
```

Final Answer: The final answer to the original input question. Please provide a structured response including the following information. If a field is not applicable, use "NA".

- Causal Effect: [The causal effect estimate]
- Method: [The method used]
- Standard Error: [The standard error of the causal effect]
- Treatment Variable: [The treatment variable]
- Outcome Variable: [The outcome variable]
- Mediator Variable: [The mediator variable, if frontdoor adjustment was used, NA otherwise]
- RCT: [True / False indicating if the data is from a randomized controlled trial, NA if not sure]
- Covariates: [List of control covariates and confounders used in the estimation model]
- Additional Variable: [Instrument, running variable, or temporal variable, if applicable]
- Results of Statistical Tests: [Key statistical results, if applicable]
- Explanation for Model Choice: [Explanation, if applicable]
- Regression Formula: [The regression formula, if applicable]

Note: Only import from the approved package list above. Do not use any other packages. Do not create any plotting.

For all outputs in code, THE 'print()' function MUST be called. If you use Action in this step, stop after generating the Action Input and wait the execution outcome from 'python_repl_ast'. If you output

the final answer in this step, do not use Action.

Here is an example of using the 'python_repl_ast':

```
Action: python_repl_ast
Action Input:
```python
# Your code goes here - only use approved packages
import pandas as pd
import numpy as np
print(df.head())
```
Begin!|
```

Question: self.query
Available causal inference methods:

- IPW (Inverse Probability Weighting): Choose the right estimand (ATE/ATT/ATC), and compute the causal effect

- Linear regression with control variables: Build a regression model with the treatment, outcome, and confounders/control variables, and compute the causal effects

- Instrumental variable: Build an instrumental variable model, and compute the causal effects associated with the treatment variable

- Matching: Choose the correct estimand (ATE/ATT/ATC), and match accordingly, and then compute the causal effects

- Difference-in-differences: Build a difference-in-differences model, and output the coefficient of the variable of interest

- Regression discontinuity design: Build a regression discontinuity design model, and output the coefficient of the variable of interest

- Linear regression / difference-in-means: Either build a regression model consisting of the treatment and outcome variables, and compute the coefficient associated with the treatment variable or compute the difference in means across treatment and control groups

- Generalized linear models / GLM: Build a GLM model, and output the coefficient of the variable of interest

- Frontdoor adjustment: Build a causal graph, identify a mediator variable between the treatment and outcome, check for frontdoor criterion, and compute the causal effect using the frontdoor adjustment formula

## F  SYNTHETIC DATA GENERATION

We use the template below to generate the context and variable labels for synthetic data.

---

**Prompt for Generating Context for Synthetic Data**

You are a helpful assistant generating realistic, domain-specific contexts for synthetic datasets.
The current dataset is designed for {method_name} studies in the domain domain.

Dataset Summary
{summary}

Previously Used Contexts (avoid duplication)
{history}

Domain-Specific Guidance
{domain_guides}

Your Tasks:

1. Propose a realistic real-world scenario that fits a `{method_name}` study in the domain `domain`. Mention whether the data was collected from a randomized trial, policy rollout, or real-world observation.

   a. Assign realistic and concise variable names in snake_case. Map original variable names like "X1" to names like "education_years".

   b. Provide a one-line natural-language description for each variable (e.g., education_years: total years of formal schooling completed by the individual.). Use newline-separated key-value format.

2. Write a paragraph describing the dataset's background: who collected it, what was studied, why, and how. Then, provide a clear description of each variable in the dataset, explaining what it represents and, where relevant, its type (e.g., continuous, binary, categorical). For binary or categorical variables, specify what the values mean.

3. Write a natural language causal question the dataset could answer. The question should:
   - Relate implicitly to the treatment and outcome
   - Avoid any statistical or causal terminology
   - Avoid naming variables directly

4. Write a 1–2 sentence summary capturing the dataset's overall intent and contents.

Return your output as a JSON object with the following keys:
- `"variable_labels":{"X1":  "education_years", ...}`
- `"description":` "<paragraph>"
- `"question":` "<causal question>"
- `"summary":` <summary>
- `"domain":` "<domain>"

Return only a valid JSON object. Do not include any markdown, explanations, or extra text.

---

**Notes on Placeholders**

`summary` provides a description of the dataset. It specifies which symbols correspond to the treatment, outcome, continuous covariates, and binary covariates. It also adds method-specific details (e.g., IV instrument, RDD cutoff, or DiD setup) and includes a statistical summary of the variables as given by df.describe().

`{history}` contains a record of previously generated dataset contexts. This is used to prevent duplication and ensure variety across generated scenarios.

`{domain_guides}` provides domain-specific guidance, such as reminding the model that education data often includes student performance and school-level features, or that healthcare data often covers treatments and recovery outcomes.

We verify that the output of the LLM does not explicitly describe the estimand or specify a causal inference method to be used. As an illustration, the following example shows a context and query generated by the LLM for an RCT dataset:

---

**Example of a Query + Context for Synthetic Data**

```
{
    "query": "Does providing housing subsidies improve the stability
    of housing situations?",

    "dataset_description": "This dataset was compiled from a
    Randomized Control Trial conducted by the Department of Housing
    and Urban Development (HUD) of the United States. The goal was
    to investigate the impact of a new housing subsidy policy on
    recipients' housing stability. Variables include the age of the
    recipient ('recipient_age'), their monthly income
    ('monthly_income'), whether they own a home ('is_homeowner',
```

```
        binary: 1 for homeowners, 0 for non-homeowners), whether they
        have dependents ('has_dependents', binary: 1 for yes, 0 for no),
        whether they reside in a rural area ('lives_in_rural_area',
        binary: 1 for rural, 0 for urban), whether they received the
        housing subsidy ('received_subsidy', binary: 1 for yes, 0 for
        no), and their self-reported housing stability
        ('housing_stability')."
    }
```

# G   ANNOTATION DETAILS

For each article we curate the following information:

- **Paper Name:** Name of the study
- **Description:** The description about the dataset that includes the collection process, purpose, and brief explanation about the variable names
- **Query:** Causal question associated with the dataset
- **Answer:** Causal effect derived in the paper
- **Standard Error:** Standard error associated with the causal effect estimate
- **Significant:** Binary variable indicating if the effect is statistically significant
- **Method:** The causal inference method
- **Treatment:** The name of the treatment variable in the dataset
- **Outcome:** The name of the outcome variable in the dataset
- **Control Covariates:** The control variables / confounders used in the estimation model
- **Interaction Variable:** The name of the variable that interacts with the treatment. This is used for measuring heterogeneous treatment effects
- **Instrument:** The variable used as an instrument. If instrumental variable is not used, this is set to null
- **Running Variable:** The running variable for Regression Discontinuity Design (RDD). If RDD is not used, we set this to null
- **Temporal Variable:** The variable denoting the timing of treatments. This is used for difference-in-differences
- **State Variable:** The variable denoting the different participating entities. This is used for two way fixed effects versions of difference in difference
- **Multi-RCT Treatment Variable:** The treatment type of interest. This is used in RCTs with multiple treatments
- **Data File:** The name of the csv file containing the data
- **Reference:** Reference to the original paper, where the result is found
- **Publication Year:** The year the original study was published
- **Domain:** The domain of the original study

# H   ADDITIONAL ANALYSIS

## H.1   BREAKING DOWN RELATIVE ERRORS BY METHOD SELECTION CORRECTNESS

To further investigate the impact of incorrect method selection on effect estimation, we compute the relative errors for examples where method selection is correct versus those where the selection is incorrect. Table 4 shows this breakdown.

| Model | Method Selection | Real | | | Synthetic | | | Textbook | | |
|---|---|---|---|---|---|---|---|---|---|---|
| | | **Error** | **%** | **Diff.** | **Error** | **%** | **Diff.** | **Error** | **%** | **Diff.** |
| Gemini-2.5-FL | Correct Method | 61.6 | 49.6 | +15.7 | 40.5 | 76.4 | +7.3 | 39.4 | 72.7 | +5.3 |
| | Incorrect Method | 77.3 | 50.4 | | 47.8 | 23.6 | | 44.7 | 27.3 | |
| Grok-4-Fast | Correct Method | 52.9 | 69.5 | +25.6 | 19.9 | 73.3 | +3.2 | 22.5 | 65.6 | +10.4 |
| | Incorrect Method | 78.5 | 30.5 | | 23.1 | 26.7 | | 32.9 | 34.4 | |
| GPT-5-mini | Correct Method | 47.9 | 71.5 | +31.6 | 7.6 | 90.5 | +23.6 | 51.5 | 70.1 | −36.4 |
| | Incorrect Method | 79.5 | 28.5 | | 31.2 | 9.5 | | 15.2 | 29.9 | |
| GPT-4o-mini | Correct Method | 67.5 | 35.1 | +9.7 | 11.0 | 23.0 | +19.6 | 26.3 | 58.2 | +15.0 |
| | Incorrect Method | 77.2 | 64.9 | | 30.5 | 77.0 | | 41.3 | 41.8 | |
| GPT-4o | Correct Method | 62.4 | 57.4 | +15.5 | 22.6 | 74.0 | +12.0 | 39.5 | 63.8 | −8.8 |
| | Incorrect Method | 77.8 | 42.6 | | 34.6 | 26.0 | | 30.7 | 36.2 | |
| OpenAI-o3 | Correct Method | 49.9 | 71.0 | +29.5 | 31.5 | 85.7 | +26.1 | 52.2 | 64.2 | −23.3 |
| | Incorrect Method | 79.4 | 29.0 | | 57.5 | 14.3 | | 28.9 | 35.8 | |
| Qw3-Next-80B-I | Correct Method | 60.2 | 57.9 | +16.1 | 32.5 | 73.9 | −5.6 | 33.2 | 77.4 | +30.9 |
| | Incorrect Method | 76.3 | 42.1 | | 26.9 | 26.1 | | 64.1 | 22.6 | |

Table 4: Impact of method selection on causal effect estimation error. Mean relative errors are averaged across all prompting strategies (Direct, CoT, PoT, ReAct). Percentage (%) indicate the proportion of examples with correct versus incorrect method selection. Diff. represents the difference in mean relative errors between correctly and incorrectly selected methods

# I  NOTES ON CAUSAL INFERENCE METHODS

This section offers a brief overview of the causal inference approaches we examine. For comprehensive theoretical foundations and detailed methodological discussions, we direct readers to textbooks on causal inference (Cunningham, 2021; Imbens & Rubin, 2015; Hernan & Robins, 2025; Peters et al., 2017).

## I.1  RANDOMIZED CONTROL TRIALS

RCTs are the gold standard for causal inference. This is because the ignorability assumption, which states that treatment assignment is independent of the potential outcomes, is satisfied by default.

$$Y(0), Y(1) \perp T \tag{2}$$

**Identification Assumption**  The key assumption is ignorability equation 2.

**Assumption Check**  Whether or not data comes from an RCT should be specified in the data description. If mentioned, we do not need to perform additional checks. We impose the assumption by design.

**Estimand**  The causal estimand of interest is Average Treatment Effect (ATE).

**Causal Estimation**  The most straightforward way to estimate causal effect is **Difference in means**. As the name states, we simply find the difference in the average outcomes for treatment and control groups. Mathematically,

$$\hat{\tau} = \sum_{i \in \text{Treatment}} \frac{1}{n_1} Y_i - \sum_{i \in \text{Control}} \frac{1}{n_0} Y_i \tag{3}$$

where $n_1$ and $n_0$ are the total number of units in treatment and control groups respectively.

In practice, we often compute $\hat{\tau}$ by regressing the outcome (Y) on treatment (T).

In some cases, the data may also contain pre-treatment covariates. These are variables measured before the experiment and are unaffected by the treatment. We often include them in our estimation to improve the precision of the causal effect measure, i.e., minimize the standard error. In such cases, the causal effect model is

$$Y = \alpha + \tau T + X\beta + \epsilon \text{ where } \epsilon \text{ is an error term uncorrelated with T and X} \qquad (4)$$

## I.2   IPW

Inverse Probability Weighting (IPW) is one of the methods for estimating causal effects from observational datasets. The key assumption underlying IPW is conditional ignorability. This states that the potential outcomes are independent of treatment assignment conditioned on confounding variables. Confounding variables are those that affect both treatment and outcome. Mathematically,

$$Y(0), Y(1) \perp T|X \qquad (5)$$

**Propensity Score** Propensity score, $e(X) \in [0,1]$, gives a measure of how likely a unit is to be treated. To estimate propensity scores, we can fit logit or probit models on the confounders X. Upon computing the propensity scores, we can directly compute IPW estimates. Note that when fitting the propensity scores, you should fit a single model for the whole data.

**Estimand** Average Treatment Effect (ATE), Average Treatment Effect on the Treated (ATT), Average Treatment Effect on the Control (ATC). The right estimand depends from problem to problem. The most popular estimand is ATT, then ATE and ATC.

**Assumption** The key assumption is conditional ignorability.

**Causal Estimation** The measures of causal effects are

$$\hat{\tau}_{ATE} = \frac{\sum_{i:T_i=1} \frac{Y_i}{e(X_i)}}{\sum_{i:T_i=1} \frac{1}{e(X_i)}} - \frac{\sum_{i:T_i=0} \frac{Y_i}{1-e(X_i)}}{\sum_{i:T_i=0} \frac{1}{1-e(X_i)}} \qquad (6)$$

$$\hat{\tau}_{ATT} = \frac{1}{n_1} \sum_{i:T_i=1} Y_i - \frac{\sum_{i:T_i=0} \frac{e(X_i)}{1-e(X_i)} Y_i}{\sum_{i:T_i=0} \frac{e(X_i)}{1-e(X_i)}} \qquad (7)$$

$$\hat{\tau}_{ATC} = \frac{\sum_{i:T_i=1} \frac{1-e(X_i)}{e(X_i)} Y_i}{\sum_{i:T_i=1} \frac{1-e(X_i)}{e(X_i)}} - \frac{1}{n_0} \sum_{i:T_i=0} Y_i \qquad (8)$$

**Assumption Check** We need to satisfy the conditional ignorability assumption. This is an untestable assumption. Experts usually use their domain knowledge to select confounding variables and justify their selection.

Another method to check is to assess the distribution of covariates in the treated and control groups. A popular method for assessing the distribution is SMD (Standardized Mean Difference). For each covariate x, we compute the standardized mean difference between treatment and control groups as:

$$\text{SMD}_x = \frac{\mu_x^{\text{treatment}} - \mu_x^{\text{control}}}{\sqrt{(\sigma_x^{\text{treatment}})^2 + (\sigma_x^{\text{control}})^2/2}}$$

where $\mu_x$ and $\sigma_x$ are the mean and standard deviation of covariate x in each group. If ignorability is approximately satisfied, the standardized mean difference should be close to zero for each confounder.

Likewise, we can also assess the propensity scores for treated and control groups. Ideally, we want the distribution of the propensity scores to be similar. If there are many confounders, we often compute the SMD for propensity scores.

Another method to assess the distribution of confounders is through visual inspection.

## I.3 MATCHING

As stated above, propensity score based estimators are highly unstable for real world problems. To improve stability, we often use matching. As the name suggests, we match each unit with its nearest neighbor. The causal effect for that particular unit is

$$\tau_i = Y_i - Y_{m_i} \quad \text{where } m_i \text{ is the unit matched to i}$$

You can think of matching as an equivalent of the nearest neighbor method in machine learning. Matching requires computing similarities. One common way to perform matching is to match units with similar propensity scores.

**Type of Matching** Just like in the nearest neighbor method, we have k-matching, i.e., for each unit, we select the K nearest neighbors, and then compute the causal effect as

$$\tau_i = Y_i - \frac{1}{K} \sum_{k=1}^{K} Y_k \tag{9}$$

Similarly, we can have matching with replacement or without replacement. Matching with replacement is more common in practice.

**Causal Estimation** You can think of matching as a preprocessing step. We compute the causal effect using the matched units. The nature of matching varies between estimands.

- **ATE** Each unit in control is matched to a unit in the treatment group and vice versa. To compute the causal effect,

$$\hat{\tau}_{ATE} = \frac{1}{N} \sum_{i=1}^{N} (Y_i - Y_{m_i}) \tag{10}$$

  Notice that we do not compute the means for treatment and control separately.

- **ATT** We only match units in the treatment group, i.e., for each unit in the treatment group, we select k nearest neighbors in the control group. The causal effect is then computed as:

$$\hat{\tau}_{ATT} = \frac{1}{n_1} \sum_{i \in \text{Treatment}} (Y_i - \frac{1}{K} \sum_{k=1}^{K} Y_{m_{i,k}}) \tag{11}$$

  where $m_{i,k}$ denotes the k-th matched control unit for treated unit i.

### I.3.1 MATCHING VS IPW

IPW is fast and relatively easier to implement. However, IPW is highly unstable when the overlap assumption is violated. Hence, in practice we often use matching. If the propensity scores are well balanced across both the control and treated units, the causal effect from matching and IPW should be similar to one another. In such cases, the estimates from IPW would be fairly reliable. However, if the balance is poor, we should use matching. Thus, you can think of matching as a preprocessing step that improves the comparability of treatment and control groups.

## I.4 DIFFERENCE IN DIFFERENCES (DID)

Difference in Differences is a quasi-experimental method for computing causal effects by addressing time-invariant confounding. For instance, suppose New Jersey raised its minimum wage, but its neighbor Pennsylvania did not. We are interested in the impact of the wage policy on employment rates across two time periods. Over time, several things could have changed that could impact employment rates, for example, tax policies. In such cases, we can use the trends in Pennsylvania to remove the effects from other time-varying factors and compute the impact of the minimum wage policy.

For DiD to be considered, we must have panel data, i.e., observations across multiple time periods (at least two). Moreover, the time-related information must not be a covariate. It must correspond to the timing of the treatments.

**Identification Assumptions** Two key assumptions underlying DiD are:

- **Parallel Trends Assumption** In the absence of treatment, the outcome trends in treatment and control groups would have evolved similarly, i.e., if there was no treatment, $E[Y_{1t} - Y_{1t-1}|\text{Treatment}] = E[Y_{0t} - Y_{0t-1}|\text{Control}]$. This states that the change in outcome in the two groups would be the same in the counterfactual scenario.
- **No Anticipatory Effects** This states that the treatment effect applies only after implementation, meaning units do not change their behavior in anticipation of future treatment.

**Assumption Check** First, we need to test if DiD is the right method. Just because the data has a time-related variable does not mean DiD is the right method. One should first identify what the treatment variable is, and then check if the time-related variable indicates the treatment timing.

The **no anticipatory effects** assumption is typically valid by design if the treatment timing is exogenous. The assumption of primary interest is the parallel trends assumption. We can test for this visually. For instance, if we have data on the outcomes for more than two periods before treatment, we can plot them and examine the slopes of the lines in the two groups. If the parallel trends assumption is valid, then the slopes should be similar, i.e., roughly parallel. In case we do not have enough pre-treatment data, say there are only two time periods (pre and post), then we can use domain knowledge to justify why parallel trends could be valid.

**Estimand** The estimand is the Average Treatment Effect on the Treated (ATT).

**Causal Estimation** There are two main scenarios under DiD:

- **Canonical DiD** This is the classical 2 period and 2 group setting. This means we have 2 time periods: pre and post treatment periods, and 2 groups: treatment and control. This applies in situations where the treatment was applied at a single point in time to a specific group. For causal effect estimation, we define two indicator variables:
  1. POST: indicator variable that is 1 if the observation is made after treatment is applied, i.e., $POST_t = 1$ if $t \geq$ treatment time
  2. TREAT: indicator variable that is 1 if unit i is in the treatment group and 0 otherwise.

  Then the causal model is:

  $$Y_{i,t} = \alpha + \beta \cdot POST_t + \gamma \cdot TREAT_i + \delta \cdot POST_t \times TREAT_i + X_{i,t}\beta + \epsilon_{i,t} \quad (12)$$

  The coefficient of interest is $\delta$, which represents the DiD estimator. We can add a valid set of control variables $X_{i,t}$. Note that these must not be affected by the treatment (bad controls).

- **Two-Way Fixed Effects (TWFE)** This is a generalized version of DiD, where the treatment is staggered, i.e., there are multiple units receiving treatment at different periods. An example of this could be the adoption of unilateral divorce laws by US states in different years. This takes the following form:

  $$Y_{i,t} = \alpha_i + \lambda_t + \delta \cdot D_{i,t} + X_{i,t}\beta + \epsilon_{i,t} \quad (13)$$

  The key coefficient of interest is $\delta$. $D_{i,t}$ is an indicator variable that is 1 if unit i has received treatment by time t. $\alpha_i$ captures unit-specific fixed effects and $\lambda_t$ captures time-specific fixed effects.

I.5 REGRESSION DISCONTINUITY DESIGN (RDD)

Regression Discontinuity Design is another quasi-experimental method. It is applicable in situations where treatment assignment is dictated by a threshold value. For instance, say we want to evaluate the impact of stimulus checks on total household spending. We could exploit the fact that stimulus checks are given to people whose annual income is less than 70k.

In RDD, the variable that determines treatment assignment is called the running variable $(r_i)$. If the cutoff is $r_0$, the treatment assignment is

$$T_i = \begin{cases} 1 & \text{if } r_i \geq r_0 \\ 0 & \text{otherwise} \end{cases} \quad (14)$$

The causal effect is thus

$$\tau_{RDD} = \lim_{r \to r_0^+} E[Y \mid R = r] - \lim_{r \to r_0^-} E[Y \mid R = r]$$

This means we are computing the causal effect around the threshold value.

**Assumption**  The identification assumption is that around the cutoff the potential outcomes are continuous. The outcomes change abruptly only when we transition from control to treatment at the cutoff.

**Assumption check**  Usually, we perform visual inspection around the cutoff. This means fitting curves to the left and right of the cutoff based on the running variable, and looking for jumps between the curves at the threshold.

### I.6  INSTRUMENTAL VARIABLES (IV)

This is another popular method for causal effect estimation. It is useful in cases with unobserved confounders. We use an instrument, which is a variable that affects the outcome only through the treatment and is independent of unobserved confounders.

IV applies to both randomized and non-randomized experiments.

- **Encouragement Design** This is the randomized case where treatment assignment is random but not all assignees accept their treatment, i.e., assignment is not the same as uptake. In such cases, we estimate the causal effect for the compliers (units who comply with their assignment). To check if this is an encouragement design, verify that the data come from a randomized experiment and that compliance information is available.
- **General IV** This describes quasi-experimental settings where we can find a variable that influences treatment but is not affected by unobserved confounders.

**Estimand**  The estimand in IV is called LATE (Local Average Treatment Effect) or CACE (Complier Average Causal Effect).

**Assumption**  The assumptions underlying IV are:

- Independence of the instrument: the instrument is independent of potential outcomes (as if random).
- Exclusion restriction and relevance: the instrument affects the outcome only through the treatment (exclusion), and it is correlated with the treatment (relevance).
- Monotonicity: the instrument moves treatment in the same direction for all units (no defiers). For instance, being selected for the draft via lottery should not cause some otherwise willing participants to avoid service while motivating others to serve.

**Assumption test**  Monotonicity and independence are usually justified by design and domain knowledge. In randomized encouragement designs, this is straightforward; otherwise, it requires a domain-based justification. To test relevance (instrument correlated with treatment), we can compute the F-statistic. The most important assumption, the exclusion restriction, is untestable and relies on substantive knowledge.

**Causal Estimation**  There are two ways to compute causal effects.

- Non-parametric: most suitable for encouragement designs.

$$\hat{\tau}_{CACE} = \frac{\dfrac{\sum_{i:Z_i=1} Y_i}{\sum_i Z_i} - \dfrac{\sum_{i:Z_i=0} Y_i}{\sum_i (1 - Z_i)}}{\dfrac{\sum_{i:Z_i=1} D_i}{\sum_i Z_i} - \dfrac{\sum_{i:Z_i=0} D_i}{\sum_i (1 - Z_i)}} \tag{15}$$

- Parametric: The classic econometric approach (2 stage least squares / 2SLS), where we first regress the treatment on the instrument, then regress the outcome on the predicted value of treatment from the first stage. Intuitively, we use the part of treatment induced by the instrument to estimate the causal effect.

## I.7 BACKDOOR AND FRONTDOOR ADJUSTMENT

The above methods focus on the potential outcomes framework. Another approach to causal inference based on Pearl's principles Pearl (2009) uses causal graphs. Causal graphs are DAGs that show how variables are causally related to one another. The general structure is $Cause \rightarrow Effect$.

For graph-based methods, the first task is to construct a causal graph, and domain knowledge comes into play here. We also draw attention to the field of causal discovery, which is mainly concerned with learning causal graphs in a principled manner from data. However, it is important to distinguish causal discovery from causal effect estimation. While causal discovery focuses on identifying causal relationships and graph structure, causal effect estimation assumes the causal structure is known and focuses on quantifying the magnitude of causal effects.

**Estimand** Backdoor-based methods can estimate ATE, ATT, or ATC. Meanwhile, frontdoor computes the ATE.

**Assumption Testing** There is no definitive external test here. Backdoor and frontdoor adjustments rely on assumptions encoded by the DAG, many of which are untestable from observational data. The validity of the results hinges on the correctness of the graph; domain knowledge is typically used to justify it.

**Link to previous methods** Matching, IPW, and backdoor adjustment are interconnected approaches. Backdoor adjustment provides a systematic framework for choose the model variables from the graph aka the adjustment set. For instance, it allows us to select the confounders that we could use for matching, propensity score computation, etc. Using this adjustment set for effect estimation justifies how the conditional ignorability assumption is met.

Hence, the focus on backdoor criterion is on identification: i.e. what variables allows us to measure causal effects in a principled manner. For estimation, we then call methods like IPW or regression on the adjustment set. For a broader discussion on the connection between potential outcomes and Pearl's framework, we refer the readers to Imbens (2020)

## J DATASET SOURCES

