# OpenReview forum: "CauSciBench: A Comprehensive Benchmark on End-to-End Causal Inference for Scientific Research"
_ICLR.cc/2026/Conference — Submitted to ICLR 2026_

### Official Review · Reviewer_CfWm · 2025-10-22

**Soundness:** 3
**Presentation:** 3
**Contribution:** 2
**Rating:** 4
**Confidence:** 4

**Summary:**

This paper introduces a novel comprehensive benchmark named CauSciBench , designed to evaluate the ability of large language models (LLMs) to perform end-to-end causal inference tasks in scientific research. The authors point out that existing causal inference benchmarks have limitations; for example, they either focus only on text-based commonsense causal reasoning or only evaluate code implementation after a method is given (like QRData), neglecting the complete analysis pipeline from a natural language problem to the final result. CauSciBench is a benchmark that evaluates the complete causal analysis pipeline , requiring models to start from a natural language description and data to autonomously complete variable selection, method choice, statistical model implementation, and result interpretation. The research results indicate that the primary reasons for model failure include a systematic bias towards using OLS , and that even when the correct method is chosen, there are still serious errors in the specific implementation.

**Strengths:**

The benchmark combines three complementary data sources: real-world research papers, synthetic data, and textbook examples. This design is very clever, as it makes it possible to not only measure the model's performance but also to diagnose the stages at which the model fails.

The paper not only reports the models' scores but also deeply analyzes their "vulnerabilities". For example, it reveals through confusion matrices that models (especially smaller ones) have an over-reliance on OLS. Furthermore, analysis of Table 4 shows that "incorrect method selection" is the primary driver of high errors. These findings are highly instructive for the future improvement of LLMs.

The paper's structure is clear, and the diagrams and experimental results are very intuitive and straightforward. The authors also provided detailed information in the appendix, including the prompt templates used , the synthetic data generation method , and the technical details of the causal inference methods involved in the evaluation. This greatly enhances the paper's authenticity and credibility.

**Weaknesses:**

1. This paper designs a benchmark model that focuses on testing the ability of Large Language Models (LLMs) to perform end-to-end causal inference tasks. I do not deny the significance of this design, but I am more concerned with how to solve these problems rather than just discovering them. Therefore, the contribution of this paper might be somewhat diminished. Perhaps the authors should have at least discussed how to use the results found in this paper to guide the correction of large language models.

2. The authors generously acknowledge the limitations of this study, which is good. However, the excessive number of limitations declared suggests that this is still a method in its preliminary stages. As the authors stated, the benchmark primarily focuses on the "Potential Outcomes Framework," mainly focuses on binary treatment scenarios, the evaluation was conducted at pass@1 (i.e., the model passes on a single attempt), etc. The method still has significant room for improvement.

3. The paper uses two main metrics: Method Selection Accuracy (MSA) and Mean Relative Error (MRE), but the measurement by these two metrics still appears rather coarse. MSA is a binary metric; it cannot distinguish between selecting a highly related but incorrect model (e.g., using IPW instead of Matching) and selecting a completely irrelevant model (e.g., using OLS on an IV problem). MRE, while focusing on the estimated value, may also "fail to fully capture the severity of the estimation failure."

**Questions:**

1. As the authors stated, the core of causal inference is not only calculating an effect value but also verifying the model's assumptions. For example, how to validate the parallel trends assumption for DiD or the instrumental variable validity for IV. It is recommended that future versions of the benchmark add this type of task.

2. Based on the results, CoT prompting is generally superior to direct prompting, but the performance of PoT and ReAct is, on the contrary, unstable. This result seems somewhat counter-intuitive. It is hoped that the authors can provide a more in-depth case study on the specific failure modes of different prompting strategies.

3. The authors mention the evaluation pipeline can "pinpointing key vulnerabilities" (Page 2), but the current analysis in Table 4 is still too coarse-grained, only providing an analysis of the accuracy of method selection. Analyses of variable accuracy, code accuracy, etc., could also be added.

4. The paper claims to evaluate "result interpretation" (in Abstract), but the evaluation metrics (MSA and MRE) seem to only focus on point estimates and methods.

---

> ### Author Response · Authors · 2025-11-23
> **Response to Reviewer CfWm**
>
> Thank you for your thorough review and recognition of our benchmark design combining three complementary data sources, the analysis of model vulnerabilities and failure modes, and the clear structure with intuitive diagrams and detailed appendix materials. We appreciate your comments and hope the following response addresses your concerns.
>
> **Response Summary** In short, (1) we performed comprehensive fine-grained pipeline analysis, (2) we clarified how our findings guide solutions for improving LLM causal reasoning, (3) we expanded evaluation metrics and provided detailed prompting strategy analysis, and (4) we clarified the notion of (statistical) interpretability in our benchmark.
>
> ---
>
> ### Weaknesses
>
> > **Concern:** I am more concerned with how to solve these problems rather than just discovering them. Perhaps the authors should have at least discussed how to use the results found in this paper to guide the correction of large language models.
>
> **Response:**
> We have added discussions on how CauSciBench facilitates solutions in the General Response section. Our fine-grained analysis pinpoints specific failure modes that guide targeted improvements: models need better strategies for identifying relevant confounders and controls from large variable sets (overlap scores are only ~50% on real data), and current models over-rely on OLS, indicating the need for more rigorous reasoning about identification assumptions and method choice. Importantly, because method and variable selection directly inform the subsequent implementation steps, our results highlight that methodological reasoning is the core bottleneck that must be addressed to improve end-to-end causal analysis.
>
> Our systematic comparison reveals why different strategies succeed or fail: CoT excels because it enables reasoning about identification assumptions; PoT focuses on implementation and is less effective for guiding reasoning about the validity of the methods; and ReAct tends to under-select controls. Likewise, across all models, we see a high tendency to default to OLS. These findings suggest concrete improvements. First, since OLS is valid for randomized trials but often inappropriate for observational data, prompts can explicitly guide models to first identify the study design, then select methods accordingly. Second, all current prompting strategies reason across the entire pipeline in one go. Breaking the process into explicit sequential steps (forcing models to complete and verify each stage before proceeding) may reduce errors by enabling more thorough reasoning for method selection.
>
>
> ---
>
> > **Concern:** The authors generously acknowledge limitations, but the excessive number suggests this is still preliminary. The benchmark primarily focuses on the Potential Outcomes Framework and binary treatment scenarios.
>
> **Response:**
> **On the Potential Outcomes Framework:** The Potential Outcomes Framework is the dominant framework for causal estimation across a range of disciplines. A wide range of techniques for effect estimation that we have covered are studied under this framework. The other approach to causal inference is Pearl's framework based on causal graphs. However, its focus is on learning causal graph structures from data. There are pre-existing benchmarks to evaluate causal discovery capabilities, some of which we have discussed in the related works section. Nevertheless, there is a connection between the two approaches. For instance, both frameworks use methods like matching and regression adjustment. The difference lies in how one chooses the confounding variables to adjust for. One exception is the frontdoor criterion, which is specific to causal graphs. To account for this, we have generated synthetic datasets from causal graphs where frontdoor adjustment can be applied.
>
> **On treatment types:** It is true that the majority of datasets in our benchmark involve binary interventions. This reflects current empirical practice: the majority of published causal studies focus on binary cases. However, we have non-binary examples, including continuous treatments and multi-treatment RCTs. One example is provided in Figure 2, which involves continuous treatment.
>
> In summary, the prominence of Potential Outcomes and binary treatments reflects mainstream practice in empirical causal research across disciplines (as shown in Fig. 3), and we do not view these as limitations of our design.
>
> **Note on pass@k:** While we recognize that pass@k evaluation is important for a more thorough analysis, we were unable to conduct these experiments due to budgetary constraints. We plan to incorporate this analysis in a future version of the study.
>
> ---

---

> > ### Author Response · Authors · 2025-11-23
> > **Response to Reviewer CfWm - Continued**
> >
> > > **Concern:** MSA is a binary metric; it cannot distinguish between selecting a highly related but incorrect model (e.g., using IPW instead of Matching) and selecting a completely irrelevant model (e.g., using OLS on an IV problem).
> >
> > **Response:**
> > It is true that the binary nature of MSA penalizes all types of mistakes equally. For this reason, we have included confusion matrix heatmaps (Figure 4) to further understand the errors in method selection, where we can observe the model's tendency to choose some methods over others. We will expand this analysis in the revised manuscript to identify patterns of misidentification, examining which types of queries lead to specific method confusion. However, we expect errors to be predominantly driven by wrong selections across fundamentally different methods rather than between similar approaches. This is because different causal inference methods involve substantial variation in model structure and underlying identification assumptions. As a result, selecting a method that is fundamentally mismatched for the given data and scenarios (e.g., using OLS for an IV problem) tends to produce substantially larger downstream errors.
> >
> > ### Questions
> >
> > > As the authors stated, the core of causal inference is not only calculating an effect value but also verifying the model's assumptions. For example, how to validate the parallel trends assumption for DiD or the instrumental variable validity for IV. It is recommended that future versions of the benchmark add this type of task.
> >
> > **Response:**
> > We agree that justifying identification assumptions is crucial to causal inference. However, assumption validation poses several challenges: assessing parallel trends in DiD typically requires visual inspection, and IV validity often relies on domain-specific reasoning to justify the exclusion restriction. While some quantitative diagnostics exist (e.g., F-test for IV), there is no systematic way to combine domain-based justification and statistical diagnostics.
> >
> > Our method selection accuracy metric partially captures this. Selecting the appropriate method requires understanding whether key assumptions are plausible given the research design. On inspection of the logs, we found that LLMs used arguments about research design and data descriptions to justify their method choices, which suggests they engage with these assumptions. Nevertheless, we acknowledge that explicitly evaluating assumption testing is an important direction for future work.
> >
> > ---
> >
> > > Based on the results, CoT prompting is generally superior to direct prompting, but the performance of PoT and ReAct is, on the contrary, unstable. This result seems somewhat counter-intuitive. It is hoped that the authors can provide a more in-depth case study on the specific failure modes of different prompting strategies.
> >
> > **Response:**
> > We provide comprehensive comparative analysis in the General Response section. CoT and PoT explicitly encode the causal analysis pipeline with step-by-step guidance, while ReAct and Direct are free-form approaches. Causal inference requires methodological reasoning and implementation, but methodological reasoning is more critical, as errors here propagate to implementation.
> >
> > CoT is particularly effective because it forces models to reason through identification assumptions, many of which are conceptual and untestable. PoT, while structured, is more implementation-focused and less effective at guiding reasoning about untestable assumptions. This explains why CoT achieves better performance, especially in method selection. ReAct shows substantially lower control overlap scores on textbook (QRData) and synthetic data, indicating it tends to select fewer controls/confounders due to its sequential processing structure. Prior work (Liu et al., 2024) has shown that PoT and ReAct excel at implementation-focused tasks, but our benchmark evaluates end-to-end causal inference from problem formulation to implementation, where methodological reasoning is more important.
> >
> > ---
> >
> > Liu, Xiao, et al. "Are LLMs Capable of Data-Based Statistical and Causal Reasoning? Benchmarking Advanced Quantitative Reasoning with Data." In Findings of the Association for Computational Linguistics: ACL 2024

---

> > > ### Author Response · Authors · 2025-11-23
> > > **Response to Reviewer CfWm - Continued**
> > >
> > > > The authors mention the evaluation pipeline can "pinpoint key vulnerabilities" (Page 2), but the current analysis in Table 4 is still too coarse-grained, only providing an analysis of the accuracy of method selection. Analyses of variable accuracy, code accuracy, etc., could also be added.
> > >
> > > **Response:**
> > > We have added comprehensive fine-grained analysis in the General Response section, revealing that models fail at two critical points: incorrect control variable selection and wrong method selection. These errors compound, as selecting the wrong controls can invalidate even correctly chosen estimation methods. Regarding code accuracy, all prompting approaches generate code for implementing the identified methods and variables. Hence, we focus on evaluating the inputs to the code: variables and methods, rather than the code itself. We add a metric that records how many attempts the model needs to produce error-free code, offering insight into how reliably the model can implement the analysis.
> > >
> > > ---
> > >
> > > > The paper claims to evaluate "result interpretation" (in Abstract), but the evaluation metrics (MSA and MRE) seem to only focus on point estimates and methods.
> > >
> > > **Response:**
> > > By "interpretation," we refer to **statistical interpretation**, specifically assessing significance through confidence intervals and p-values. If zero lies in the 95% confidence interval, the effect is not statistically significant at the 0.05 level; otherwise, it is significant. This practice is common across disciplines. Hence, in many papers, statistical significance is marked with asterisks. Once these statistics are available, LLMs can readily interpret them. Therefore, we focus on getting the important statistics correct, and thus include information on standard errors that allows the computation of p-values and the assessment of whether the result is significant or not.
> > >
> > > From the analysis of logs, we verified that LLMs correctly interpret the direction/magnitude of effects with respect to statistical significance. Nevertheless, we acknowledge that we currently focus on statistical interpretation and do not currently evaluate aspects like assumption discussion, limitation acknowledgment, or causal claim qualification with respect to identification challenges, as these involve combining both qualitative and quantitative arguments. Currently, our benchmark focuses on the quantitative aspect.
> > >
> > > ---
> > >
> > > We sincerely appreciate your suggestions and constructive feedback. In sum, we have added further explanations on comprehensive fine-grained analysis, demonstration of how findings guide solutions, expanded metrics, and clarified how our benchmark supports statistical interpretability. We hope these clarifications sufficiently address your concerns.

---

> ### Author Response · Authors · 2025-11-26
> **Kind Reminder for Author-Reviewer Discussion**
>
> Dear Reviewer CfWm,
>
> We warmly remind you that the author-reviewer discussion phase will end soon. If our response has helped addressing some of your concern, we kindly ask you to consider raising your score, we're very happy to provide further information.

---

### Official Review · Reviewer_Picg · 2025-10-28

**Soundness:** 1
**Presentation:** 2
**Contribution:** 2
**Rating:** 2
**Confidence:** 4

**Summary:**

The paper construct a benchmark aiming for end-to-end causal analysis for scientific research. The data sources covers real-world research paper, synthetic framework, and textbook datasets.
It employs two matrices: Method Selection Accuracy (MSA) and Mean Relative Error (MRE). In the experiment section, 8 LLMs are evaluated in the above three settings.

**Strengths:**

- It focuses on the causal inference task that is essential to scientific research.
- The proposed benchmark has more broad coverage on data sources and scientific disciplines.
- The empirical results reveals the difficulty and complexity of automating causal inference tasks with LLMs.

**Weaknesses:**

- It needs more elaboration on the unique challenges behind *end-to-end causal analysis*. Based on the line 53~80, my best understanding of *end-to-end causal analysis* is the pipeline of selecting both *variables* and *method*, and these are highly related to the data preprocessing and model specifications that covered in the existing benchmarks.
- It needs more evaluation matrices to match the stated benchmarking target. Currently, there are only two matrices: Method Selection Accuracy (MSA) and Mean Relative Error (MRE). There are some key steps stated by the paper but not evaluated, for example: "formulation of problems from natural language descriptions", "choice of treatment/effect/confounders", and “result interpretation”. I suggest to employ more advanced evaluating method like LLM-as-a-judge to enrich the set of matrices.
- It needs more diverse querying templates mimicking both expert and non-expert users. The queries are already well formulated causal questions. For example, in figure 2, the query is "What is the effect of education on earnings?" This query has already stated the expected variables. It is not surprising that LLMs can select correct data columns in the .csv files with variable description.
- Implicite data pre-processing in the csv files. Take the example in the figure 2. Some pre-processing is from the original dataset by the experts, like *log of the wage* and *indicator of locations*; another type of pre-processing is from incomplete variable list. For example, variables about family structure (*daded*, *nodaded*, *momed*, and *nomomed*, ..) are not presented. There are about 50 variables in the original dataset [1], while figure 2 only displays 8 variables. If the remaining variables are omitted due to page limitation, it would be much better to provide these information in appendix; If the remaining variables are filtered, then what is the filtering criteria? and how would this influence the difficulty of the tasks?
- Insufficient discussion on contamination. Although the contamination concern is discussed in section 3 and limitation section. It is unclear how such potential contamination can influence the evaluation results and interpretation. This uncertainty weaken the practical utility of the proposed benchmark.

[1] Using Geographic Variation in College Proximity to Estimate the Return to Schooling

**Questions:**

Please refer to the concerns stated in the previous section. And also the following questions:
 - In Table 2, textbook datasets has the best method selection accuracy. Is this a signal of perceivable data contamination that memorizing the textbook's method choice? How can we interpret such results?
 - In Table 4, when using incorrect methods, why LLMs are more likely to give lower error in the textbook group? The current explanation is "both methods yield similar results" (line 407). I failed to find out why this could explain the large difference in mean relative errors. I suggest more detailed discussion to avoid potential misunderstanding.

---

> ### Author Response · Authors · 2025-11-23
> **Response to Reviewer Picg**
>
> Thank you for your thorough review and for recognizing the benchmark's focus on causal inference tasks, an important component of scientific research, the broad coverage of data sources and disciplines, and the illustration of challenges in automating causal inference with LLMs. We hope the clarifications below adequately address the concerns and questions raised.
>
> **Response Summary** In short, (1) we performed a comprehensive fine-grained pipeline analysis with additional metrics, (2) we provided clarifying information on data preprocessing and details on the dataset sources, (3) we conducted systematic contamination analysis with empirical validation, and (4) we clarified query diversity and evaluation scope. Accordingly, we will also revise our manuscript.
>
> ---
>
> ### Weaknesses
>
> > **Concern:** It needs more elaboration on the unique challenges behind end-to-end causal analysis. My best understanding is the pipeline of selecting both variables and methods, which are highly related to data preprocessing and model specifications that are covered in existing benchmarks.
>
> **Response:**
> Current benchmarks on causal estimation focus on evaluating LLM's ability to implement specified models and/or variables. Related work (Liu et al., 2024) only considers the numerical value (i.e., the causal effect) and does not annotate quantities like standard deviation, which are important for statistical interpretation of results.
>
> In contrast, our benchmark does not hint at which causal model to implement. Rather, we pose a query relevant to the dataset and have the model perform suitable causal analysis whose results could answer it. This involves choosing treatment, outcome, methods, and computing the related effect. We have now added comprehensive fine-grained analysis on the end-to-end causal reasoning process in the General Response. These additional metrics provide a complete evaluation of the key components of the causal inference pipeline.
>
> ### Reference
> Liu, Xiao, et al. "Are LLMs Capable of Data-Based Statistical and Causal Reasoning? Benchmarking Advanced Quantitative Reasoning with Data." In Findings of the Association for Computational Linguistics: ACL 2024
>
> ---
>
> > **Concern:** Currently, there are only two metrics: MSA and MRE. Key steps like "choice of treatment/effect/confounders" and "result interpretation" are not evaluated. I suggest employing more advanced evaluating methods like LLM-as-a-judge.
>
> **Response:**
> We want to clarify that when we say "formulation of problems from natural language query," we mean creating a causal inference problem altogether by identifying treatment, outcome, control variables, and selecting the method. We will modify the manuscript to make this more explicit.
>
> We have now included fine-grained analysis beyond MSA and MRE: Treatment Selection Accuracy, Outcome Selection Accuracy, Control Overlap Score, and Number of Attempts to succeed. These metrics directly assess the key steps mentioned: choice of treatment/outcome/confounders/controls. The results are presented and discussed in the General Response section.
>
> Employing LLM-as-a-judge would be a valid approach for evaluating qualitative aspects like result interpretation. However, this would involve external evaluation not directly supported by our benchmark framework. Currently, our focus is on quantifiable aspects of causal inference that can be objectively evaluated against reference values. We recognize that evaluation of LLM's reasoning is important work and is a natural direction for future work.
>
> ---
>
> > **Concern:** The queries are already well formulated causal questions. For example, in Figure 2, the query is "What is the effect of education on earnings?" This query has already stated the expected variables. It is not surprising that LLMs can select correct data columns.
>
> **Response:**
> Figure 2 shows only a sample query. We use diverse phrasings across our benchmark. Some sample queries include "Did the marketing push help increase in-app purchases?" and "Is the Self-treatment method effective in increasing voter turnout?" While our queries must specify the causal relationship for evaluation purposes, we vary how explicitly variables are referenced. For the query "How effective is the Neighbors scheme in increasing voter turnout?", the dataset contains no column labels mentioning "Neighbors" or "voter turnout." This requires LLMs to identify correct variables from context beyond keyword matching.
>
> Variable naming varies across datasets, affecting task difficulty. Some datasets have explicit names like "treatment," while others require inference from context. Our query formulation is largely dictated by the specific model or finding in the reference study. For fair comparison, we must specify which quantities' cause-and-effect relationships we are interested in; otherwise, many cause-effect pairs could emerge within the same dataset, making fair comparison impossible.
>
> ---

---

> ### Author Response · Authors · 2025-11-23
> **Response to Reviewer Picg - Continued**
>
> > **Concern:** Take the example in Figure 2. There are about 50 variables in the original dataset, while Figure 2 only displays 8 variables. What is the filtering criteria? How would this influence task difficulty?
>
> **Response:**
> The example in Figure 2 was obtained from the 'causaldata' package in R [2], which is available under an MIT license. The full version of the original dataset is also available, but we could not find information about its usage license [1]. Hence, we used the 'causaldata' version instead.
>
> In general, whenever possible, we use the full data version shared by the original authors as is. However, when licenses are not available or unclear, we use alternative versions (which may be simplified) that have clear usage licenses. Approximately 85% of our datasets are the versions released by the original authors.
>
> When datasets do not contain full documentation of their variables, finding exact definitions becomes challenging, particularly for examples with large number of variables. In such cases, we retain the original dataset (some licenses do not allow sharing derivatives), but in the description, we include definitions only for the variables we are certain of. Moreover, many of the variables whose definitions we are uncertain of are redundant and do not convey relevant information on the causal model.
>
> In the updated manuscript, we will add details about the publications and data sources for all real-world tasks in the Appendix while indicating whether any of the datasets are simplified versions of the original.
>
> [1] David Card, "Using Geographic Variation in College Proximity to Estimate the Return to Schooling," NBER Working Paper 4483 (1993)
>
> [2] Huntington-Klein, Nick. 'causaldata: Example Data Sets for Causal Inference Textbooks', 2024.
>
> ---
>
> > **Concern:** Insufficient discussion on contamination. It is unclear how such potential contamination can influence the evaluation results and interpretation. This uncertainty weakens the practical utility of the proposed benchmark.
>
> **Response:**
> We conducted systematic contamination assessment and empirical validation experiments by rewriting each dataset's narrative and variable names, updating all columns accordingly, and rerunning the full ReAct pipeline. We also ran a source-identification test, asking the model to recall the originating paper. The tables below show the results for 39 queries in the real and QRdata datasets:
>
> | Dataset Version  | Model        | Accuracy (%) | Relative Error | Source Identification (lower is better) |
> |------------------|--------------|--------------|----------------|------------------------------------------|
> | Real             | GPT-4o-mini  | 24.32        | 59.64          | 11/39                                    |
> | Perturbed Real   | GPT-4o-mini  | 22.22        | 49.53          | 1/39                                     |
> | Real             | GPT-4o       | 67.57        | 50.56          | 12/39                                    |
> | Perturbed Real   | GPT-4o       | 53.12        | 62.69          | 0/39                                     |
>
> | Version           | Model        | Accuracy (%) | Relative Error | Source Identification (lower is better) |
> |-------------------|--------------|--------------|----------------|------------------------------------------|
> | Original QRData   | GPT-4o-mini  | 56.76        | 26.67          | 19/39                                    |
> | Perturbed QRData  | GPT-4o-mini  | 56.76        | 33.84          | 0/39                                     |
> | Original QRData   | GPT-4o       | 50.00        | 42.45          | 9/39                                     |
> | Perturbed QRData  | GPT-4o       | 52.94        | 33.97          | 0/39                                     |
>
> Across both QRData and real datasets, accuracy and relative error remain largely stable under perturbation, while source identification drops to near zero, indicating the models are not relying on memorized sources. The nonzero source identification scores in the original setting represent hallucinated citations rather than genuine recall. The stability of model performance under perturbation, combined with the elimination of source citations, provides strong evidence that memorization is not driving benchmark results.

---

> > ### Author Response · Authors · 2025-11-23
> > **Response to Reviewer Picg - Continued**
> >
> > ### Questions
> >
> > > In Table 2, textbook datasets have the best method selection accuracy. Is this a signal of perceivable data contamination that is memorizing the textbook's method choice?
> >
> > **Response:**
> > There are two plausible reasons for the higher method selection accuracy on textbook datasets. First, data contamination could be a factor. Textbooks (QRData) use seminal datasets like the Lalonde data. It is highly likely that these datasets were part of the LLM training data. Second, textbook descriptions are simpler and more pedagogical, making analysis easier.
> >
> > However, our contamination analysis suggests memorization is not the primary driver. When we perturbed textbook datasets, method selection accuracy remained stable. For GPT-4o, the difference in method selection accuracy between perturbed and non-perturbed versions is only 2.94%. If memorization were dominant, accuracy should drop significantly under perturbation. Given the small performance change, we conclude that the simplified nature of textbook datasets is the primary driver of higher accuracy rather than memorization.
> >
> > ---
> >
> > > In Table 4, when using incorrect methods, why are LLMs more likely to give lower error in the textbook group? The current explanation is "both methods yield similar results" (line 407). How does this explain the large difference in mean relative errors?
> >
> > **Response:**
> > In the textbook data, the majority of examples with incorrect method selection were related to the IHDP data. Since the data comes from a randomized trial, the preferred method would be difference-in-means or regression adjustment. However, the LLM tends to select matching/inverse probability weighting (IPW) instead. Because the data is randomized, even when matching methods or IPW are used, the causal effect estimates remain relatively stable and do not vary significantly from the true answer. Hence, the overall error tends to be small.
> >
> > In contrast, for observational datasets, errors are larger because they are non-randomized and more sensitive to method choice. Wrong method selection leads to substantial bias because proper confounder adjustment becomes critical. This demonstrates that method selection strongly predicts estimation quality in observational settings, even though multiple methods may converge in randomized settings. We will clarify this distinction in the revised manuscript to avoid potential misunderstanding.
> >
> > ---
> >
> > We sincerely appreciate your detailed feedback and questions. We have addressed your points on fine-grained analysis, data preprocessing, contamination checks, and evaluation metrics in the clarifications above, and we hope these explanations adequately address your concerns.

---

> > > ### Comment · Reviewer_Picg · 2025-11-27
> > >
> > > I appreciate the authors’ efforts in addressing my previous concerns. The revision improves clarity in several sections.
> > >
> > > Here are my remaining concerns and comments.
> > >
> > > ------
> > >
> > >  **1. About the elaboration on the “unique challenges”**
> > >
> > > The explanation remains unclear to me.
> > >
> > > - The authors state that _“our benchmark does not hint at which causal model to implement.”_ However, the referenced paper also does not reveal or specify the underlying causal model, so the distinction remains ambiguous.
> > >
> > > - The authors further mention that the referenced paper _“does not annotate quantities like standard deviation.”_ I was only able to locate a brief discussion of standard deviation in the appendix of that work. If this is one of the major methodological differences the current submission wishes to highlight, stronger justification and a clearer explanation in the main paper would be helpful. Currently, the contrast remains under-motivated.
> > >
> > > **2. Additional metrics**
> > >
> > > The Fine-Grained Analysis looks nice to me.
> > >
> > > The low 'Control Variable Overlap' performance of the ReAct method is interesting.
> > >
> > > Could the authors clarify **what actions the ReAct agent is permitted to take**?
> > >
> > > For instance, if the agent were allowed to run lightweight code (e.g., correlation checks among variables), one might anticipate substantially higher performance. Clarifying the action space is important for interpreting these results.
> > >
> > > **3. Formulated causal questions**
> > >
> > > Thanks for your clarification. From the perspective of potential users of this benchmark, one may want to know how many variants are utilized to reconstruct the questions (like variable renaming). The rationale behind each variant and its ratios.
> > >
> > >  **4. The filtered variable set**
> > >
> > > Have you compared the results between "original datasets with licenses" and "causaldata"? If the results differ significantly, it may be better to drop the "causaldata" subset.
> > >
> > > **5. Data contamination analysis**
> > >
> > > Thanks for sharing the updated results — the perturbation experiment is very interesting, especially the sharp decrease in _Source Identification rate_.
> > >
> > > - Could you further elaborate on the conclusion that _“the high source identification scores in the original setting represent hallucinated citations rather than genuine recall.”_ I am not an expert in the research about hallucination.
> > >
> > > - From the perspective of accuracy, the variation in the metrics seems stable. It would be much better if the additional metrics in the Fine-Grained Analysis could also be provided.

---

> > > > ### Author Response · Authors · 2025-12-03
> > > > **Response to Reviewer Picg (Round 2)**
> > > >
> > > > Thank you for your follow-up questions and comments. Below, we provide additional clarifications addressing each one of them.
> > > >
> > > > > **Concern 1:** About the elaboration on the "unique challenges"
> > > >
> > > > **Response:**
> > > >
> > > > For reference, here is an example from QRData:
> > > > ```json
> > > > {"data_description": "A study is conducted to measure the effect of a marketing push on user engagement, specifically in-app purchases. Some customers who were assigned to receive the push are not receiving it because they likely have an older phone that doesn't support the type of push the marketing team designed. The dataset app_engagement_push.csv contains records for 10,000 random customers. Each record includes whether an in-app purchase was made (in_app_purchase), if a marketing push was assigned to the user (push_assigned), and if the marketing push was successfully delivered (push_delivered).",
> > > >   "question": "What is the Local Average Treatment Effect (LATE) of receiving the marketing push on in-app purchases, as estimated using linear regression and instrumental variable, rounded to two decimal places?",
> > > >   "answer": "3.29"}
> > > > ```
> > > >
> > > > The question explicitly specifies the inference method to use: instrumental variables, along with the causal effect measure of interest: LATE (causal estimand).
> > > >
> > > > Meanwhile, we annotate the same example as:
> > > > ```json
> > > > {"description": "Same as the original",
> > > >   "question": "Did the marketing push help increase in-app purchases?",
> > > >   "method": "iv",
> > > >   "effect": 3.29,
> > > >   "std_error": 0.7165,
> > > >   "is_significant": 1,
> > > >   "treatment_var": "push_delivered",
> > > >   "outcome_var": "in_app_purchase",
> > > >   "instrument_var": "push_assigned"}
> > > > ```
> > > >
> > > > Based on the description, we expect the LLM to realize that the push was randomly assigned, but not all assignments were delivered. This is a classic example of encouragement design. Thus, we expect the LLM to realize this, select instrumental variable (IV) as the inference method, implement it appropriately, and compute the associated causal effect and its statistical significance level.
> > > >
> > > > In general, QRData examples explicitly refer to the inference method and/or causal estimand [Average Treatment Effect (ATE), Average Treatment Effect on the Treated (ATT), LATE, etc.] and/or model variable names. By variable names, we mean the exact names by which they appear in the dataset, not the attributes they represent. Similarly, by telling the model what method or estimand to consider, we are not assessing its ability to reason about the description and select the appropriate inference method. Therefore, we do not mention these explicitly in our queries and expect the LLM to identify them based on the input information.
> > > >
> > > > Additionally, QRData annotations include only the causal effect value. We annotate treatment, outcome, and control variables, standard error, statistical significance, and model-specific variables (for example, instruments for IV). Annotations of the model variables at a more granular level (e.g., treatment, outcome) allow us to evaluate performance across different parts of the pipeline. Likewise, the explicit annotation of standard errors provides a measure of uncertainty associated with the estimate and allows us to compute the statistical significance of results.
> > > >
> > > > The above comments are with respect to causal estimation problems (also referred to as causal inference in the statistics community), which is our focus. QRData also contains queries for causal discovery (learning causal graphs from data), statistical analysis, and text-based causal reasoning. Those are not the focus of our benchmark.
> > > >
> > > > ---
> > > >
> > > > > **Concern 2: Additional metrics**
> > > >
> > > > **Response:**
> > > >
> > > > The ReAct agent can execute Python commands via python_repl_ast. Similarly, it has access to the following libraries: pandas, numpy, scipy, scikit-learn, statsmodels, dowhy, linearmodels, rdd, and econml. The agent is allowed to generate code, execute it, and then plan its next step accordingly. This process continues until it settles on a final answer.
> > > >
> > > > ReAct is more free-form, unlike CoT, where we explicitly guide the model through the reasoning steps. Nevertheless, the logs for ReAct show that when performing the analysis, models think about the need to adjust for confounders or to include control variables. However, we believe the lack of explicit guidance prevents it from engaging as rigorously as in CoT.
> > > >
> > > > ---

---

> > > > > ### Author Response · Authors · 2025-12-03
> > > > > **Response to Reviewer Picg (Round 2) - Continued**
> > > > >
> > > > > > **3. Concern: Formulated causal questions**
> > > > >
> > > > > **Response:**
> > > > >
> > > > > To clarify, we do not rename the variables in the dataset. We use the naming as in the source. Likewise, in the queries, we do not refer to the names verbatim. They describe the causal quantities of interest, and we expect the LLM to map these to appropriate variables in the dataset.
> > > > >
> > > > > When we refer to query variants, we mean the phrasing of the causal question. For instance, we could ask: What is the effect of X on Y? Does X increase Y? or Does X have any effect on Y?, where X and Y are quantities / attributes of interest.
> > > > >
> > > > > While the phrasings are different, they require the same core causal analysis: identification of treatment/outcome (mapping of X and Y to dataset variables), method selection, model-specific variables, and finally implementation of the effects. They vary in how one interprets the findings to answer the final query. The first question focuses on the magnitude, the second on the sign (and the statistical uncertainty), and the third on whether there is a plausible effect or not. However, for all the queries, the core task of causal analysis remains the same. Hence, we did not impose a fixed rule regarding the phrasing structure.
> > > > >
> > > > > ---
> > > > >
> > > > > > **4. Concern: The filtered variable set**
> > > > >
> > > > > **Response:**
> > > > >
> > > > > We have not performed a close analysis between the results for datasets in causaldata and those for datasets made available in their original versions by the authors. Datasets from CausalData constitute a small fraction of our real-dataset (< 15%). Similarly, for some studies, we do not have access to the original data. Hence, data from the package serves as an alternative. Likewise, for some examples, such as the castle data, the data is relatively more complex. More importantly, we are able to replicate the results of the reference analysis (or get close to it) using the dataset.
> > > > >
> > > > > While the datasets may be relatively more simplified, we believe their strengths (described above) outweigh this limitation. Moreover, we believe the nature of the datasets helps increase the diversity of our benchmark in terms of task difficulty. Therefore, we believe including the datasets will be helpful.
> > > > >
> > > > > ---
> > > > >
> > > > > > **5. Concern: Data contamination analysis**
> > > > >
> > > > > **Response:**
> > > > >
> > > > > "Hallucinated citations rather than genuine recall" means the LLM references the wrong papers as the dataset source. In our experiment, we asked the LLM to identify the reference study from which the example query and dataset were possibly drawn from. Source identification scores denote the number of examples for which the LLM responded. For example, a score of 5 means that the LLM responded to 5 examples. We manually checked the LLM responses to assess if it recalled the correct paper. On inspection, we found that most of them were incorrect citations. GPT-4o-mini has a high source identification score, but this does not mean the citations are correct. It simply means the model responded more frequently, not that those responses are accurate.
> > > > >
> > > > > The tables below provide comparison across the pipeline components: treatment and outcome accuracy, control overlap, and number of attempts to generate error-free code.
> > > > >
> > > > > ## Treatment Identification Accuracy (%)
> > > > >
> > > > > | Model | Prompting Strategy | Real | Perturbed Real | Original QRData | Perturbed QRData |
> > > > > |-------|-------------------|------|----------------|-----------------|------------------|
> > > > > | GPT-4o-mini | React | 86.956 | 88.323 | 88.235 | 85.507 |
> > > > > | GPT-4o | React | 73.913 | 75.897 | 91.176 | 92.345 |
> > > > >
> > > > > ## Outcome Identification Accuracy (%)
> > > > >
> > > > > | Model | Prompting Strategy | Real | Perturbed Real | Original QRData | Perturbed QRData |
> > > > > |-------|-------------------|------|----------------|-----------------|------------------|
> > > > > | GPT-4o-mini | React | 94.444 | 76.470 | 94.594 | 91.891 |
> > > > > | GPT-4o | React | 94.594 | 82.758 | 92.105 | 97.05 |
> > > > >
> > > > > ## Control Variable Overlap (%)
> > > > >
> > > > > | Model | Prompting Strategy | Real | Perturbed Real | Original QRData | Perturbed QRData |
> > > > > |-------|-------------------|------|----------------|-----------------|------------------|
> > > > > | GPT-4o-mini | React | 80.787 | 56.428 | 92.669 | 66.5333 |
> > > > > | GPT-4o | React | 57.106 | 35.952 | 89.657 | 70.5333 |
> > > > >
> > > > > ## Average Number of Code Generation Attempts
> > > > >
> > > > > | Model | Prompting Strategy | Real | Perturbed Real | Original QRData | Perturbed QRData |
> > > > > |-------|-------------------|------|----------------|-----------------|------------------|
> > > > > | GPT-4o-mini | React | 2 | 1.9 | 1.625 | 1.51 |
> > > > > | GPT-4o | React | 1.5 | 1.684 | 1.294 | 1.3 |

---

> ### Author Response · Authors · 2025-11-26
> **Kind Reminder for Author-Reviewer Discussion**
>
> Dear Reviewer Picg,
>
> We'd like to warmly remind you that the author-reviewer discussion phase will end soon. If our response has helped addressing some of your concern, we kindly ask you to consider raising your score, we're very happy to provide further clarification if needed

---

### Official Review · Reviewer_3iLr · 2025-10-31

**Soundness:** 3
**Presentation:** 3
**Contribution:** 3
**Rating:** 8
**Confidence:** 5

**Summary:**

The paper introduces a large-scale benchmark designed to evaluate an LLM's ability to perform end-to-end causal inference in realistic scientific contexts. In contrast to prior benchmarks that focus on specific causal inference queries, CauSciBench focuses an entire causal analysis pipeline, including natural language problem formulation, variable selection, methodological choice, model implementation, and interpretation of estimates.

Contributions:

* A benchmark that requires models to go through an end-to-end inference processs -- identify treatment, outcome, and confounding variables, choose appropriate identification strategies, implement them in Python, and interpret results.
* Integrates  research papers across several scientific domains, supplemented with synthetic datasets generated.
* Introduces quantitative metrics called Method Selection Accuracy (MSA) and Mean Relative Error (MRE), and analyzes model errors across steps of the causal pipeline.
* Comparative study of six leading LLM models under multiple prompting paradigms (Direct, Chain-of-Thought, Program-of-Thought, and ReAct). Results show that the best system a mean relative error high enough to cast doubt on LLM ability to do end-to-end causal inference.
* The benchmark reveals systematic overreliance on OLS, cascading implementation errors, and limited transfer from synthetic or textbook data to real research data.

**Strengths:**

The paper is original in framing causal inference as a full end-to-end capability for LLMs, as opposed to focusing on sub-tasks. The quality of the benchmark construction is high, combining real-world research data, textbook examples, and synthetic cases. The clarity of presentation is strong—methodology, examples, and evaluation setup are clearly described, with well-structured figures and prompt templates help with reproducibility. The significance lies in establishing a standardized way to measure whether LLMs can autonomously perform scientific causal analysis.

**Weaknesses:**

* The evaluation pipeline may not fully capture partial reasoning competence, e.g., correctly identifying treatment/outcome but failing in implementation.
* The contextualization of failures could be deepened—for instance, more analysis of why models default to OLS or how CoT reasoning collapses under noise.

**Questions:**

How exactly are the “ground truth” causal effects verified for the real-world paper-derived tasks? Are they always replicated numerically in Python, or are some drawn from published tables without independent validation? Clarifying this would help assess the reliability and reproducibility of the benchmark’s reference values.

You mention two rounds of expert validation for causal queries—could you provide inter-rater agreement?

---

> ### Author Response · Authors · 2025-11-23
> **Response to Reviewer 3iLr**
>
> Thank you for your thoughtful review and strong endorsement with a rating of 8. We are encouraged by your recognition of the originality in assessing the end-to-end causal inference capabilities of LLMs, the quality of benchmark construction, the comprehensive presentation with reproducible prompt templates, and the establishment of a framework for measuring the abilities of LLMs to perform causal analysis. We hope the following clarifications address your remaining concerns.
>
> ---
>
> ### Weaknesses
>
> > **Concern:** The evaluation pipeline may not fully capture partial reasoning competence, e.g., correctly identifying treatment/outcome but failing in implementation.
>
> **Response:**
>
> We have performed comprehensive fine-grained analysis, specifically examining treatment and outcome variable identification accuracy and overlap assessment of control/confounder variables. The results demonstrate that while models can identify primary causal variables with reasonable success (average treatment accuracy of 70.7% and outcome accuracy of 80.1% across all models on the real data), they struggle substantially with control/confounder selection (an average of 51.5% overlap on real datasets), revealing a critical bottleneck. The specific results and broader analysis are described in the General Response section.
>
> ---
>
> > **Concern:** The contextualization of failures could be deepened—for instance, more analysis of why models default to OLS or how CoT reasoning collapses under noise.
>
> **Response:**
>
> **Why models default to OLS:** OLS is the simplest model to use and does not require extensive reasoning or preprocessing compared to other methods. Likewise, OLS tends to be the default baseline model in empirical papers. Hence, due to simplicity and high prevalence, LLMs, especially smaller ones, tend to select OLS more often. This observation aligns with findings in Zhang et al. [1], who point out that LLMs struggle with tasks requiring advanced causal reasoning.
>
> **Why CoT reasoning degrades:** Based on our evaluation of model outputs, we observe that CoT struggles when datasets have variables with similar names and research designs with multiple outcomes of interest. While CoT can reason about causal identification from data descriptions, this reasoning becomes more challenging as complexity increases. In these cases, models more frequently default to OLS, suggesting that when reasoning about assumptions underlying the methods becomes difficult, models fall back to simpler, more common methods. This explains both the prevalence of OLS across all dataset types and lower performance on queries with more complex descriptions and variable naming.
>
> Our comparison of prompting strategies in the General Response provides a detailed analysis of these failure modes across different dataset types.
>
> [1] Zhang, Cheng, et al. 2023. "Understanding Causality with Large Language Models: Feasibility and Opportunities." arXiv preprint arXiv:2304.05524.
>
> ---
>
> ### Questions
>
> > How exactly are the "ground truth" causal effects verified for the real-world paper-derived tasks? Are they always replicated numerically in Python, or are some drawn from published tables without independent validation?
>
> **Response:**
>
> The ground truth reference causal effects are obtained from published studies. We replicate all analyses in Python to ensure the results are reproducible. Many studies use R or Stata, and not all libraries have exact Python equivalents (for example, `matchit` in R). Due to variations in how packages work, there could be minor differences. If discrepancies >5% occur, we omit the query. For synthetic tasks, ground truth is computed directly from a known data-generating process. For textbook tasks, we verify the causal effects match textbook answers and derive additional information, such as standard deviations, using standard methods. We will add these details to the updated manuscript.
>
> ---
>
> > You mention two rounds of expert validation for causal queries—could you provide inter-rater agreement?
>
> **Response:**
>
> We would like to clarify that our "two rounds of expert validation" did not require measuring agreement between reviewers, because the task involved objective factual checks rather than subjective judgment. In each round, experts verified that every query (1) was factually consistent with the original study and (2) contained no information leakage, i.e., did not provide hints on the causal inference method to use.
>
> ---
>
> We sincerely appreciate your strong endorsement and valuable feedback, and we hope our responses have provided additional clarifications.
>
> ---

---

### Official Review · Reviewer_F88Z · 2025-11-01

**Soundness:** 3
**Presentation:** 2
**Contribution:** 2
**Rating:** 6
**Confidence:** 3

**Summary:**

The paper introduces CauSciBench, a benchmark for evaluating large language models (LLMs) on end-to-end causal inference tasks grounded in real scientific research. It claims to assess models across the entire causal inference pipeline—from identifying treatment and outcome variables to implementing and interpreting statistical methods—using real, synthetic, and textbook-based datasets. The results show that even leading models such as OpenAI-o3, have high mean relative errors, implying a large performance gap in research-level causal reasoning.

**Strengths:**

- Overall, this paper provides a comprehensive evaluation framework for assessing Causal Inference agents in realistic scenarios, effectively complementing various elements that prior works lacked. To perform causal inference as an agent, the system should be able to execute everything from variable selection to modeling independently, which this framework evaluates.
- Additionally, unlike previous works that relied entirely on synthetic data, this framework includes both real-world scenario developed from research paper, seems to be more extensive than relying only textbook, and synthetic scenarios, which enhances its completeness as an evaluation framework.
- Particularly, the aspect of providing freedom in method selection and implementation within the model is intriguing. While prior works dealt with frameworks limited to specific methods, this approach evaluates whether model implementation is possible without method constraints, making it a scalable framework in terms of LLM reasoning and tool implementation. It appears appropriate for assessing the reasoning and agentic abilities of LLMs that will continue to improve. For example, in Source 2, removing direct constraints on specific methods that prior works imposed and giving models freedom seems to be a reasonable setting for scalably evaluating an agent's reasoning ability.

**Weaknesses:**

- When generating LLM-generated data, there is a need to introduce filtering or fixation logic to resolve or mitigate hallucination problems, which seems to be absent in this work. The authors claim that for synthetically generated data (Data source 2), they created data scenarios corresponding to synthetic causal effects in Eq (1) through GPT-4o prompting, but it is unclear how they detect, filter, or fix low-quality data.
- Regarding evaluation metrics, I'm not sure if it's necessary to introduce MSA separately in addition to MRE. First, in the definition of MSA, one has to question whether it's optimal design that a candidate method approaching the performance of a reference method receives the same evaluation as a candidate method that doesn't. Even if a method other than the reference method is used, isn't MRE itself what ultimately matters? For a comprehensive analysis, introducing additional different metrics would have been more appropriate. How would one explain a situation where a "wrong method" is selected but achieves a low MRE?
- The interpretive discussion of model behavior remains broad and descriptive, without deeper causal analysis of why specific prompting strategies or architectures succeed or fail across datasets. The study lacks granular failure categorization, providing limited insight into the precise failure modes such as variable misidentification, code logic errors, or assumption mismatches.

**Questions:**

- It appears that information obtained during real-world dataset generation could be utilized to improve synthetic scenarios. What do you think about this? For example, the authors curate causal queries reviewed by 2 experts in data source 1, which could be used as few-shot examples for synthetic data generation to improve the quality of synthetic data (for instance, to reduce the probability of hallucinated data). Besides this, there seems to be room to apply techniques commonly used in general LLM synthetic data generation.
- As the authors mentioned, when evaluating LLMs in the academic domain, analysis related to data contamination is necessary, which requires reproducibility in terms of data generation. In lines 203-204, although the authors stated that the main focus of this paper is "introducing this dataset," with a bit more effort, it seems this could function not only as a generated dataset but also as an evaluation dataset generation framework. What do you think about this possibility?
- Looking at Appendix F, it states that previously used contexts are incorporated to ensure diversity, but it appears as though the entire history utilized during synthetic data generation is concatenated in the prompt. Is this correct? This approach seems inappropriate as it would be limited in terms of context length and long instruction following once the dataset size to be generated exceeds a certain threshold (perhaps several tens of scenarios would already be enough to hamper instruction following). However, if the paper's main focus is on the generated dataset itself rather than data generation algorithms, I don't consider this a major weakness, which is why I've framed this as a question rather than a weakness.
- The authors stated that the main focus of this paper is the generated dataset itself rather than the data generation algorithm. However, while there is a visual overview of data generation in the main body figures, there doesn't seem to be a visual overview of how LLMs are evaluated using the generated dataset. Given that the authors appear to be highlighting differentiated evaluation elements compared to prior worksas their main contribution, shouldn't a visualization of how the generated dataset is used for evaluation be a priority?

**Details Of Ethics Concerns:**

None.

---

> ### Author Response · Authors · 2025-11-23
> **Response to Reviewer F88Z**
>
> Thank you for your thoughtful review and acknowledgment of the comprehensive evaluation framework, the inclusion of real-world research papers alongside synthetic and textbook data, and the benchmark's focus on evaluating causal inference capabilities without prescribing any specific method. We hope the following responses address your concerns.
>
> **Response Summary** In short, (1) we implemented rigorous filtering and validation procedures for synthetic data generation, (2) we expanded evaluation metrics with fine-grained pipeline analysis, and (3) we have explained the importance of method selection accuracy.
>
> ---
> ### Weaknesses
>
> > **Concern:** Filtering or fixation logic to resolve or mitigate hallucination problems, it is unclear how they detect, filter, or fix low-quality data.
>
> **Response:**
> We agree that filtering LLM-generated outputs is essential, and we have done so. The final version is a *curated subset* of all generated scenarios. We manually inspected all outputs and removed those with major formatting issues, unwanted mentions of causal methods, or missing variable definitions. Fixable issues (e.g., variable type mismatches, such as calling binary data continuous) were corrected manually.
>
> ---
>
> > **Concern:** In the definition of MSA, one has to question whether it's optimal design that a candidate method approaching the performance of a reference method receives the same evaluation as a candidate method that doesn't. Even if a method other than the reference method is used, isn't MRE itself what ultimately matters? How would one explain a situation where a "wrong method" is selected but achieves a low MRE?
>
> **Response:**
> Method choice in causal inference cannot be judged solely by numerical accuracy. One challenge with causal inference in real-world settings is that there is no single ground truth method or metric to optimize for. Practitioners choose methods by combining domain knowledge with research design, arguing why their chosen method can reliably measure the causal effect in the given setting. Hence, MSA assesses whether a conceptually appropriate method was chosen (where ground truth is based on our curated set of peer-reviewed papers).
>
> Regarding the second point, an incorrect method can still achieve a low MRE for statistical reasons, such as less noisy data or when multiple methods converge in simple settings. We observe similar behavior in synthetic data settings, where data being less noisy and complex generally yields lower error despite wrong method selection. On the other hand, for real-world scenarios, method selection strongly affects estimation quality.
>
> ---
>
> > **Concern:** The interpretive discussion of model behavior remains broad and descriptive, without deeper causal analysis of why specific prompting strategies or architectures succeed or fail across datasets.
>
> **Response:**
> Please refer to the General Response for our comprehensive fine-grained analysis results and detailed comparison of prompting strategies and their failure modes across datasets.
>
> ---
>
> ### Questions
>
> > Information obtained during real-world dataset generation could be utilized to improve synthetic scenarios. For example, the authors curate causal queries reviewed by 2 experts in data source 1, which could be used as few-shot examples for synthetic data generation.
>
> **Response:**
> For this work, we chose to separate them to allow independent evaluation of the two collections, where one source does not influence the other. In our current approach, we address data quality concerns differently. We generated the numerical data first from fixed causal model specifications, then asked an LLM to add contexts and labels to the generated dataset. Because we generated the data from specified causal models and also informed the LLM about this, the queries were consistent with their contexts and the underlying causal structure.
>
> Regarding applying common LLM-based synthetic data generation techniques, these approaches typically generate data using an input dataset as a reference [Nguyen et al. (2024)]. As mentioned above, we avoided this because we wanted to keep the three scenarios independent from each other. Likewise, we also could not generate data from scratch in a more open-ended manner, where we prompt the LLM with the context and let it produce the data [Long et al. (2024)]. We needed fixed model specifications and known causal effects to establish ground truth, so this approach was not suitable.
>
> #### References
> Long, L., Wang, R., Xiao, R., Zhao, J., Ding, X., Chen, G., et al. On LLMs-driven synthetic data generation, curation, and evaluation. Findings of the Association for Computational Linguistics: ACL 2024, 11065–11082.
>
> Nguyen, D., Gupta, S., Do, K., Nguyen, T., and  Venkatesh, S). Generating realistic tabular data with large language models. arXiv preprint arXiv:2410.21717, 2024.
>
>
> ---

---

> ### Author Response · Authors · 2025-11-23
> **Response to Reviewer F88Z - Continued**
>
> > With a bit more effort, it seems this could function not only as a generated dataset but also as an evaluation dataset generation framework. What do you think about this possibility?
>
> **Response:**
> There are several challenges to making this a more general framework. First, dataset creation required significant manual effort. We had to review papers and materials to assess suitability. We avoided papers proposing highly specific estimators to ensure general-purpose libraries could be used. Second, some papers employ specific analyses like constructing novel instruments from existing variables. Evaluating these in an automated manner is challenging. For now, our current focus is foundational: evaluating whether LLMs can work effectively with existing variables. Third, while we used notebook LLM to compile dataset information from papers, they made numerous errors requiring manual correction. Many publications lacked variable label documentation, which we had to curate ourselves.
>
> The synthetic data generation component could potentially be automated by using LLMs to generate numerical data and corresponding contexts. However, LLMs still frequently make errors when generating scenarios. While using an additional LLM to verify and correct errors is possible, this would significantly increase costs, a major constraint.
>
> ---
>
> > It appears as though the entire history utilized during synthetic data generation is concatenated in the prompt. This approach seems inappropriate as it would be limited in terms of context length and long instruction following.
>
> **Response:**
> We only use a brief summary of the context (the `"summary"` JSON field), not the full scenario descriptions. Similarly, we do not generate all examples at once; we generate data in batches of 50 per method. The history of previously used contexts is reset when generating data for each method. The batch size was chosen considering the scalability of manual review. The total context length is around 100 sentences, so this should not pose context length or instruction following issues.
>
> We appreciate your detailed reading of our manuscript and will add clarification in our appendix to minimize possible confusion.
>
> ---
>
> > While there is a visual overview of data generation, there doesn't seem to be a visual overview of how LLMs are evaluated using the generated dataset. Shouldn't a visualization of how the generated dataset is used for evaluation be a priority?
>
> **Response:**
> We agree that a visualization of the evaluation process would enhance clarity. Accordingly, we will add a figure describing the following workflow:
>
>  Dataset + Query → LLM → Outputs (Method, Effect, Variables) → Evaluation → Metrics, where Metrics include Method Accuracy, Effect Error, and Variable Overlap.
>
> Namely, the LLM receives the input data (CSV file), its description, and a query of interest (e.g., "Does an increase in X affect Y?"). It then performs a suitable causal analysis to answer the query and outputs the estimated causal effect, standard error, the method used, and the statistical interpretation of the result. Based on these outputs, we compute method accuracy, error in causal effect magnitude, and other evaluation metrics.
>
> ---
>
> We sincerely appreciate your suggestions and feedback. We hope these clarifications sufficiently address your concerns. If you have any further questions or require additional clarification, we would be glad to provide it.

---

### Author Response · Authors · 2025-11-23
**General Response to All Reviewers**

We sincerely thank all reviewers for their valuable suggestions and constructive feedback. We are encouraged by the recognition that end-to-end causal inference is critical for scientific reasoning and that CauSciBench addresses an important gap in evaluating LLM capabilities for scientific research. Several reviewers appreciated the high quality of our benchmark construction, combining real-world research papers with synthetic and textbook data, as well as the comprehensive evaluation framework. We have carefully addressed all concerns and described changes we will make in our revised manuscript. Below we present responses to two major concerns shared by multiple reviewers: (1) fine-grained analysis of the complete causal inference pipeline, and (2) comparative analysis of different prompting strategies, and how the findings can help improve LLM's abilities to perform causal inference. We provide more tailored responses to each reviewer in the individual response sections.

---

## 1. Fine-Grained Analysis

We performed comprehensive fine-grained analysis to evaluate the complete causal inference pipeline:

- **Variable Identification**: Models must identify which variable represents the treatment (cause) and which represents the outcome (effect). We evaluate this through treatment and outcome selection accuracy.
- **Method Selection**: Models must choose an appropriate statistical method to measure the causal effect. We assess this through method selection accuracy.
- **Control Variable/Confounders Selection**: Models must identify control or confounding variables that need to be accounted for. We evaluate this through control overlap scores.
- **Effect Estimation**: Models implement the chosen method and compute the causal effect. We assess this through mean relative error.

Additionally, we report the **average number of attempts** required for each model to generate error-free code to implement the chosen inference method. Each model was allowed up to 3 retry attempts.

We present results on GPT models below and will add complete results in our camera-ready version.

---

## Treatment Variable Identification Accuracy (%)

| Model | Prompting Strategy | Real-World | Textbook | Synthetic |
|-------|---------|------------|----------|-----------|
| GPT-4o-mini | Direct | 69.75 | 82.86 | 99.19 |
| | CoT | 67.23 | 81.82 | 98.37 |
| | PoT | 68.03 | 85.29 | 95.83 |
| | ReAct | 67.23 | 82.35 | 94.31 |
| GPT-4o | Direct | 70.40 | 82.86 | 99.19 |
| | CoT | 71.54 | 82.86 | 99.19 |
| | PoT | 72.22 | 82.86 | 100.00 |
| | ReAct | 77.59 | 82.86 | 97.56 |
| GPT-5-mini | Direct | 70.97 | 82.86 | 97.54 |
| | CoT | 73.33 | 81.82 | 98.36 |
| | PoT | 71.77 | 88.57 | 100.00 |
| | ReAct | 68.87 | 83.87 | 98.23 |

---

## Outcome Variable Identification Accuracy (%)

| Model | Prompting Strategy | Real-World | Textbook | Synthetic |
|-------|---------|------------|----------|-----------|
| GPT-4o-mini | Direct | 75.86 | 89.74 | 99.30 |
| | CoT | 82.56 | 91.89 | 100.00 |
| | PoT | 75.43 | 86.84 | 100.00 |
| | ReAct | 78.24 | 91.89 | 99.30 |
| GPT-4o | Direct | 82.22 | 92.31 | 100.00 |
| | CoT | 82.02 | 92.31 | 100.00 |
| | PoT | 82.22 | 87.18 | 100.00 |
| | ReAct | 83.63 | 92.31 | 100.00 |
| GPT-5-mini | Direct | 79.33 | 92.31 | 100.00 |
| | CoT | 80.00 | 91.89 | 99.30 |
| | PoT | 79.66 | 92.31 | 98.56 |
| | ReAct | 80.26 | 91.18 | 98.46 |

---

## Control Variable Overlap (%)

| Model | Prompting Strategy | Real-World | Textbook | Synthetic |
|-------|---------|------------|----------|-----------|
| GPT-4o-mini | Direct | 49.36 | 96.33 | 98.95 |
| | CoT | 47.54 | 95.58 | 98.94 |
| | PoT | 50.03 | 96.38 | 98.61 |
| | ReAct | 47.66 | 94.31 | 96.00 |
| GPT-4o | Direct | 51.36 | 96.91 | 76.80 |
| | CoT | 50.35 | 96.80 | 78.22 |
| | PoT | 54.26 | 96.91 | 91.63 |
| | ReAct | 40.31 | 90.38 | 69.36 |
| GPT-5-mini | Direct | 50.16 | 97.01 | 94.14 |
| | CoT | 53.69 | 96.30 | 95.64 |
| | PoT | 62.12 | 96.67 | 98.76 |
| | ReAct | 55.28 | 96.91 | 94.50 |

---

## Average Number of Code Generation Attempts

| Model | Prompting Strategy | Real-World | Textbook | Synthetic |
|-------|---------|------------|----------|-----------|
| GPT-4o-mini | Direct | 2.56 | 2.80 | 2.94 |
| | CoT | 2.57 | 2.77 | 2.97 |
| | PoT | 2.27 | 2.57 | 2.92 |
| | ReAct | 1.08 | 1.18 | 1.06 |
| GPT-4o | Direct | 2.31 | 2.67 | 2.21 |
| | CoT | 2.21 | 2.66 | 2.22 |
| | PoT | 2.26 | 2.83 | 2.54 |
| | ReAct | 1.38 | 1.50 | 1.31 |
| GPT-5-mini | Direct | 2.56 | 2.90 | 2.78 |
| | CoT | 2.56 | 2.84 | 2.84 |
| | PoT | 2.56 | 2.76 | 2.74 |
| | ReAct | 1.98 | 1.94 | 2.29 |

---

> ### Author Response · Authors · 2025-11-23
> **General Response to All Reviewers - Continued**
>
> ### Key Findings
> The results reveal failures in both variable identification and control/confounder selection. While outcome and treatment identification achieve relatively high accuracy, the overlap scores for model covariates tend to be lower across all dataset types, especially the real dataset. This disparity suggests that models can identify the primary causal variables with reasonable success but struggle to determine the appropriate set of controls and confounders needed for valid causal estimation. One key reason for this difficulty is the large number of variables present in the real datasets, which makes it challenging for models to systematically evaluate and select the relevant subset.
>
> **In summary, models fail particularly at two key stages: incorrect control variable selection and wrong method selection.** These errors compound, as selecting the wrong controls can invalidate even correctly chosen estimation methods, especially for observational data.
>
> ---
>
> ## 2. How Our Benchmark Advances Causal Reasoning
>
> ### Comparison of Prompting Strategies
>
> CoT and PoT explicitly encode the causal analysis pipeline with step-by-step guidance for variable identification, method selection, and implementation. ReAct and Direct are free-form approaches without explicit guidance.
>
> Causal inference requires methodological reasoning (identifying variables, selecting controls, choosing methods) and implementation (executing the chosen method). Methodological reasoning is more critical, as errors here propagate to implementation. CoT is particularly effective because it forces models to reason through identification assumptions, many of which are conceptual and untestable. PoT, while structured, is more implementation-focused and less effective at guiding reasoning about untestable assumptions. This explains why CoT achieves better performance, especially in terms of method selection.
>
> Prior work (Liu et al., 2024) has shown that PoT and ReAct excel at implementation-focused tasks. However, our benchmark evaluates end-to-end causal inference from problem formulation to implementation, where methodological reasoning is more important. In this setting, CoT's explicit reasoning guidance proves more effective.
>
> ReAct shows substantially lower control overlap scores on textbook (QRData) and synthetic data compared to other methods. These datasets use most variables in the dataset as controls in the estimation models, especially synthetic datasets. From this observation, it is evident that ReAct tends to select fewer controls/confounders in the estimation model. Additionally, ReAct required the fewest attempts to successfully terminate across all datasets. This stems from its sequential processing structure. Unlike other prompts that perform everything in one go, ReAct proceeds one step at a time.
>
> ### Implications for LLM Research
>
> Given the above findings, we believe that guided prompting approaches like CoT and PoT benefit causal inference. These methods are particularly useful for reasoning-stages like method selection. While ReAct offers computational efficiency through sequential processing, its tendency to under-select controls/confounders makes it unsuitable for causal inference tasks where proper covariate adjustment is important. Future work could explore hybrid approaches that combine the methodological guidance of CoT with the computational efficiency of sequential processing methods like ReAct.
>
> ---
>
> ## Summary of Additional Clarifications
>
> We have made substantial clarifications addressing all reviewer concerns, including data quality validation and filtering, expanded evaluation metrics beyond MSA and MRE, systematic contamination analysis across real and textbook datasets, and clarification of the scope of interpretation evaluation. Detailed discussions are provided in our individual responses to each reviewer.
>
> —
>
> Liu, Xiao, et al. "Are LLMs Capable of Data-Based Statistical and Causal Reasoning? Benchmarking Advanced Quantitative Reasoning with Data." In Findings of the Association for Computational Linguistics: ACL 2024

---

### Author Response · Authors · 2025-12-03
**Rebuttal Summary for AC**

We sincerely appreciate AC's hard work in light of this extraordinary circumstance and provide a concise summary of rebuttal to make things easier.

We are encouraged by reviewers' broad recognition of CauSciBench's contributions as the first **end-to-end causal reasoning benchmark grounded in real-world peer-reviewed scientific papers across various discipline**

* **Grounded in Scientific Publications with Broad Coverage**: The focus on real-world research publications rather than synthetic data provides more extensive and realistic evaluation scenarios. (Reviewer F88Z)
* **Inform Future Model Development by Fine-Grained Analysis**: The evaluation pipeline that identifies specific stages where LLMs struggle, particularly in method identification and variable selection, offering actionable insights to guide future model development for causal analysis. (Reviewer CfWm)
* **Clarity of Structure and Presentation**: The clear methodology, intuitive diagrams, comprehensive experimental setup, and detailed appendix materials, including prompt templates and technical specifications. (Reviewers F88Z, 3iLr, CfWm)

While the core design and contributions remain unchanged, we have performed fine-grained analyses, systematic contamination validation, and expanded evaluation metrics. Altogether, these changes will improve the clarity of the paper and make the findings more insightful.

We are delighted to receive a strong recommendation from Reviewer 3iLr with a rating of 8. Furthermore, Reviewer F88Z maintained their positive assessment at a rating of 6. We remain confident that all raised concerns have been addressed through our explanations despite the incident that made 2 reviewers' unable to wrap up discussion with us.

## Summary of Revisions
We addressed all reviewer concerns with additional analysis. This involved running additional experiments and analyses to strengthen the evaluation framework and provide insights into model failures. Likewise, we have clarified details about our methodology and explained the rationale behind our approach.

### Core Improvements and Explanations
**C1. Fine-grained pipeline analysis to identify specific bottlenecks across the causal inference process.** (Reviewers F88Z, 3iLr, CfWm, Picg)

[**Revision 1**]: We performed a fine-grained analysis across the causal inference pipeline, including treatment/outcome identification accuracy, control variable overlap, and number of code-generation attempts. From the results, we identified that the two major bottlenecks are method selection and control variable selection.

**C2. Analysis of prompting strategies and their effectiveness across different evaluation dimensions.** (Reviewers F88Z, CfWm)

[**Revision 2**]: We conducted a detailed analysis of the prompting strategies. We found that CoT performs particularly well by explicitly guiding reasoning steps for methodological assumptions; ReAct excels in code generation but shows lower performance in control variable selection. Accordingly, we explained why these findings differ from related work focused on implementation tasks. Our benchmark is more challenging in that it requires LLMs to reason about method and variable selection.

**C3. Data contamination concerns.** (Reviewers Picg)

[**Revision 3**]: We conducted a contamination analysis through perturbation experiments on "Real" and "QRData" subsets. Namely, we perturbed dataset descriptions and variable names and ran source identification tests for both perturbed and non-perturbed versions. Method selection errors were very high for both versions, while source identification dropped to near-zero in perturbed cases, demonstrating no clear signal of contamination.

**C4. Expansion of interpretability evaluation and evaluation metrics.** (Reviewers 3iLr, CfWm, Picg)

[**Revision 4**]: We clarified that interpretability in our context refers to statistical interpretability based on confidence intervals and p-values. To enable assessment based on these quantities, we annotate both the statistical significance and standard errors. The latter can be used together with the causal effect values to compute confidence intervals and p-values. Additionally, we expanded evaluation metrics beyond MSA and MRE to include granular assessment of treatment/outcome identification, control variable overlap, and implementation success rates.

**C5. Data preprocessing transparency and dataset documentation.** (Reviewer Picg)

[**Revision 5**]: We clarified our approach for compiling datasets, filtering criteria, and formulating causal questions. Approximately 85% of datasets use the original author-released versions, while the remaining 15% are simplified versions of the original. We used the simplified versions because they are available under the MIT license, and thus can be shared easily.

---

### Meta-Review · Area_Chair_2QbF · 2026-01-05

**Summary:**

This paper introduces CauSciBench, a benchmark for evaluating end-to-end causal inference capabilities of LLMs grounded in real scientific studies, complemented by textbook and synthetic datasets.

Reviewers broadly agreed that the benchmark is well-motivated, carefully constructed, and fills a clear gap left by prior causal reasoning benchmarks that focus on isolated subtasks.
Key concerns centered on metric coverage, data contamination, query formulation, and whether the benchmark sufficiently captures the full causal reasoning pipeline.
The rebuttal added substantial fine-grained analysis, contamination validation, and clarified the scope and interpretation of the benchmark, resolving many of the initial concerns.

**Reviewer Concerns:**

Addressed by rebuttal:
- Pipeline coverage: The authors added fine-grained metrics for treatment/outcome identification, control overlap, and code-generation attempts, directly addressing requests to evaluate intermediate reasoning stages.
- Contamination: Systematic perturbation experiments and source-identification tests provide credible evidence that results are not driven by memorization.
- Metric justification: The role of MSA vs. MRE is clarified, with confusion matrices and analysis explaining why method choice matters even when numerical error is low.
- Prompting analysis: Detailed comparison of CoT, PoT, and ReAct clarifies why CoT performs better in causal reasoning tasks.
- Dataset transparency: Clarifications on preprocessing, licensing constraints, and use of simplified datasets were provided.

Still outstanding:
- Residual conceptual disagreement: One reviewer remains unconvinced that the benchmark’s distinction from prior work is sufficiently sharp, despite added explanations.
- Scope limitations: The benchmark focuses on statistical causal estimation (potential outcomes, mostly binary treatments) and does not yet evaluate assumption validation or qualitative interpretation, which the authors acknowledge as future work.

**Reviewer Scores:**

Estimated score changes after discussion:
- 3iLr: 8 → 8 (strong accept, unchanged).
- F88Z: 6 → 6 (positive but cautious, unchanged).
- CfWm: 4 → 4–5 (concerns largely addressed, but contribution seen as more diagnostic than solution-oriented).
- Picg: 2 → 2–3 (some concerns alleviated, but fundamental skepticism remains).

---

### Decision · Program_Chairs · 2026-01-26

Reject